# Preventing erosion of X-chromosome inactivation in human embryonic stem cells

Marissa Cloutier[1,9], Surinder Kumar [1,8,9], Emily Buttigieg[1,9], Laura Keller[2,3,4,5], Brandon Lee[1], Aaron Williams[1], Sandra Mojica-Perez[2,3,4,5], Indri Erliandri[2,3,4,5], Andre Monteiro Da Rocha[2,3,4,5,6], Kenneth Cadigan[7], Gary D. Smith[2,3,4,5] & Sundeep Kalantry [1✉]

X-chromosome inactivation is a paradigm of epigenetic transcriptional regulation. Female human embryonic stem cells (hESCs) often undergo erosion of X-inactivation upon prolonged culture. Here, we investigate the sources of X-inactivation instability by deriving new primed pluripotent hESC lines. We find that culture media composition dramatically influenced the expression of XIST lncRNA, a key regulator of X-inactivation. hESCs cultured in a defined xenofree medium stably maintained XIST RNA expression and coating, whereas hESCs cultured in the widely used mTeSR1 medium lost XIST RNA expression. We pinpointed lithium chloride in mTeSR1 as a cause of XIST RNA loss. The addition of lithium chloride or inhibitors of GSK-3 proteins that are targeted by lithium to the defined hESC culture medium impeded XIST RNA expression. GSK-3 inhibition in differentiating female mouse embryonic stem cells and epiblast stem cells also resulted in a loss of XIST RNA expression. Together, these data may reconcile observed variations in X-inactivation in hESCs and inform the faithful culture of pluripotent stem cells.

[1] Department of Human Genetics, University of Michigan Medical School, Ann Arbor, MI 48109, USA. [2] Department of Molecular & Integrative Physiology, University of Michigan Medical School, Ann Arbor, MI 48109, USA. [3] Department of Obstetrics & Gynecology, University of Michigan Medical School, Ann Arbor, MI 48109, USA. [4] Department of Urology, University of Michigan Medical School, Ann Arbor, MI 48109, USA. [5] Department of Physiology, University of Michigan Medical School, Ann Arbor, MI 48109, USA. [6] Department of Internal Medicine & Cardiology, University of Michigan Medical School, Ann Arbor, MI 48109, USA. [7] Department of Molecular, Cellular, and Developmental Biology, University of Michigan Medical School, Ann Arbor, MI 48109, USA. [8] Present address: Department of Pathology, University of Michigan Medical School, Ann Arbor, MI 48109, USA. [9] These authors contributed equally: Marissa Cloutier, Surinder Kumar, Emily Buttigieg. ✉email: kalantry@umich.edu

Human pluripotent stem cells (hPSCs) offer the possibility to model early human development in vitro and are substrates for regenerative medicine[1-3]. The promise of hPSCs relies on their ability to faithfully maintain their epigenetic and transcriptional profiles in culture.

X-chromosome inactivation is a paradigm of epigenetic transcriptional regulation that equalizes X-linked gene expression between female and male mammals[4-6]. Once inactivated, with a few key exceptions, replicated copies of the silenced X chromosome remain stably inactive in descendant cells[7]. X-inactivation is an experimentally tractable system to interrogate epigenetic transcriptional regulation because two equivalent X chromosomes become transcriptionally divergent and these divergent transcriptional states are subsequently stably transmitted across mitotic cell division.

X-inactivation has been studied extensively in mouse embryos and mouse embryonic stem cells (mESCs). In the female preimplantation mouse embryo, all cells undergo imprinted inactivation of the paternal X chromosome[8-10]. At the peri-implantation stage, the inactivated paternal X chromosome is reactivated in the pluripotent epiblast progenitor cells[11,12]. Conventionally cultured mESCs capture this transient population of pluripotent cells and harbor two active X chromosomes[13]. Upon differentiation, pluripotent mouse embryonic epiblast cells as well as mESCs inactivate either the maternal or the paternal X chromosome in individual cells in a process termed random X-inactivation[13-18].

In contrast to the mouse, human female preimplantation embryos do not undergo imprinted inactivation of the paternal X chromosome[19-22]. Instead, both X chromosomes in female preimplantation human embryos appear to initiate some degree of silencing[19-22]. The X chromosome-encoded long noncoding XIST RNA, whose expression is a hallmark of the inactive X chromosome and is required for stable X-inactivation in mice[10,23], is expressed from and coats in *cis* both X chromosomes in most cells of female human blastocyst-stage embryos[19-21]. XIST RNA coating in turn recruits a diverse array of proteins to the X chromosome that silence gene expression[24-28]. Despite XIST RNA coating of the X chromosomes, however, X-linked genes are not fully silenced in preimplantation human embryos[19-21].

Unlike female mESCs, which harbor two activeXs[13], female human ESC (hESC) lines display variable patterns of X-inactivation[22,29-49]. This variability appears to reflect both differences in X-inactivation patterns in early mouse vs. human female embryos and potentially how hESCs are derived and cultured.

The pattern of X-inactivation in early preimplantation human embryos is partially recapitulated in vitro through the derivation of 'naïve' pluripotent female hESCs[22,46-50]. In these naïve hESC lines, a proportion of cells display XIST RNA coating of both X chromosomes but do not appear to transcriptionally inactivate the XIST RNA-coated Xs, similar to cells in early female human embryos. Most female hESCs cultured in naïve conditions, however, harbor one XIST RNA-coated X chromosome that is transcriptionally active[22,46,48-50]. The heterogeneity of XIST RNA expression in naïve female hESCs appears to be due to the coexistence in culture of at least two populations of pluripotent cells[47]. Blocking autocrine basic fiborblast growth factor (bFGF) signaling reduces this heterogeneity and is reported to yield nearly all hESCs with two XIST RNA-coated X chromosomes[47], recapitulating the pattern observed in epiblast cells of preimplantation female human embryos[20,21]. Differentiation of these naïve hESCs into the 'primed' pluripotent hESCs results in a majority of the cells undergoing random X-inactivation and exhibiting XIST RNA coating of a single X chromosome that is transcriptionally inactive[47].

Compared to naïve hESCs, primed pluripotent hESCs capture a later stage of embryonic development and may be analogous to mouse epiblast stem cells (mEpiSCs)[3,51-56]. Female mEpiSCs contain one inactivated, XIST RNA-coated X chromosome[14]. Primed female hESCs, by contrast, exhibit at least three patterns of X-inactivation. Primed female hESCs can harbor no inactive X chromosome, one inactive-X, or a leaky inactive-X[29-43,45]. Upon prolonged culture, many primed female hESC lines lose XIST RNA coating, which is accompanied by the expression of a subset of previously silenced genes from the inactive X chromosome[29,31,35,36,38,39,43,45,57,58]. This loss of XIST RNA coating and re-expression of silenced X-linked genes has been termed X-inactivation erosion[31,34,35,38,42,43,58]. Loss of XIST RNA coating and the leaky expression of inactive X-linked genes also characterize cultured human induced pluripotent stem cells (hiPSCs) and are thought to be irreversible[35,36,42,45,57,59]. Increased X-linked gene expression due to X-inactivation erosion or failure can be deleterious to development and differentiation[23,45,60]. The instability of the epigenetically inactivated X chromosome lends caution to the use of pluripotent female human cells in disease modeling and regenerative medicine.

The underlying reasons for XIST RNA loss and X-inactivation instability in cultured hESCs are unclear. To gain insights into changes in X-inactivation in hESCs, we derived new hESC lines and analyzed expression of XIST RNA and other X-linked genes in these hESCs under diverse culture conditions and across many passages. Our results implicate the presence of lithium chloride or other GSK-3 inhibitors in the culture media as a cause of XIST RNA expression loss in female hESCs.

## Results

**Loss of XIST RNA coating in cultured hESCs.** To test the kinetics of X-inactivation in hESCs, we derived and characterized a series of new female hESC lines under primed pluripotency conditions on human fibroblast feeder cells (HFFs) (Fig. 1a; Supplementary Fig. 1; Methods). Upon passaging via mechanical splitting of the hESC colonies, we performed RNA fluorescence in situ hybridization (FISH) to capture XIST RNA coating at each passage in these cells. RNA FISH permits visualization of XIST RNA coating and nascent X-linked gene expression in nuclei of individual cells[61]. To minimize bias due to clonal expansion of the mechanically passaged hESCs, we quantified and stratified the percentage of nuclei with XIST RNA coats on a per colony basis with a minimum of 100 nuclei counted per colony.

After derivation on HFFs, we passaged the first hESC line analyzed, UM33-4, onto a Matrigel-coated surface and grew the cells at atmospheric oxygen concentration (20%) in mTeSR1 medium. Matrigel is a widely used extracellular matrix substrate that bypasses the need for feeder cells in culturing hESCs and mTeSR media are commonly used to culture hESCs on Matrigel[62-66]. In the UM33-4 hESCs, we quantified the percentage of nuclei with XIST RNA coats per colony at each passage (P) from P1 to P29 (Fig. 1b–d; Source Data file). From P1-7, all nuclei in all colonies harbored single XIST RNA coats. From P8 onwards, however, the percentage of XIST RNA-coated nuclei per colony significantly decreased (general linear model, $p = 0.002$) and at P27 XIST RNA coating disappeared altogether in all nuclei in all colonies. During subsequent passaging, hESC line UM33-4 stably maintained the complete absence of nuclei with XIST RNA coats. Of note, nuclei lacking XIST RNA coats were devoid of any XIST RNA FISH signals, suggesting that a lack of XIST RNA coating reflected an absence of XIST RNA expression since the RNA FISH assay detects stabilized as well as nascent RNAs[61].

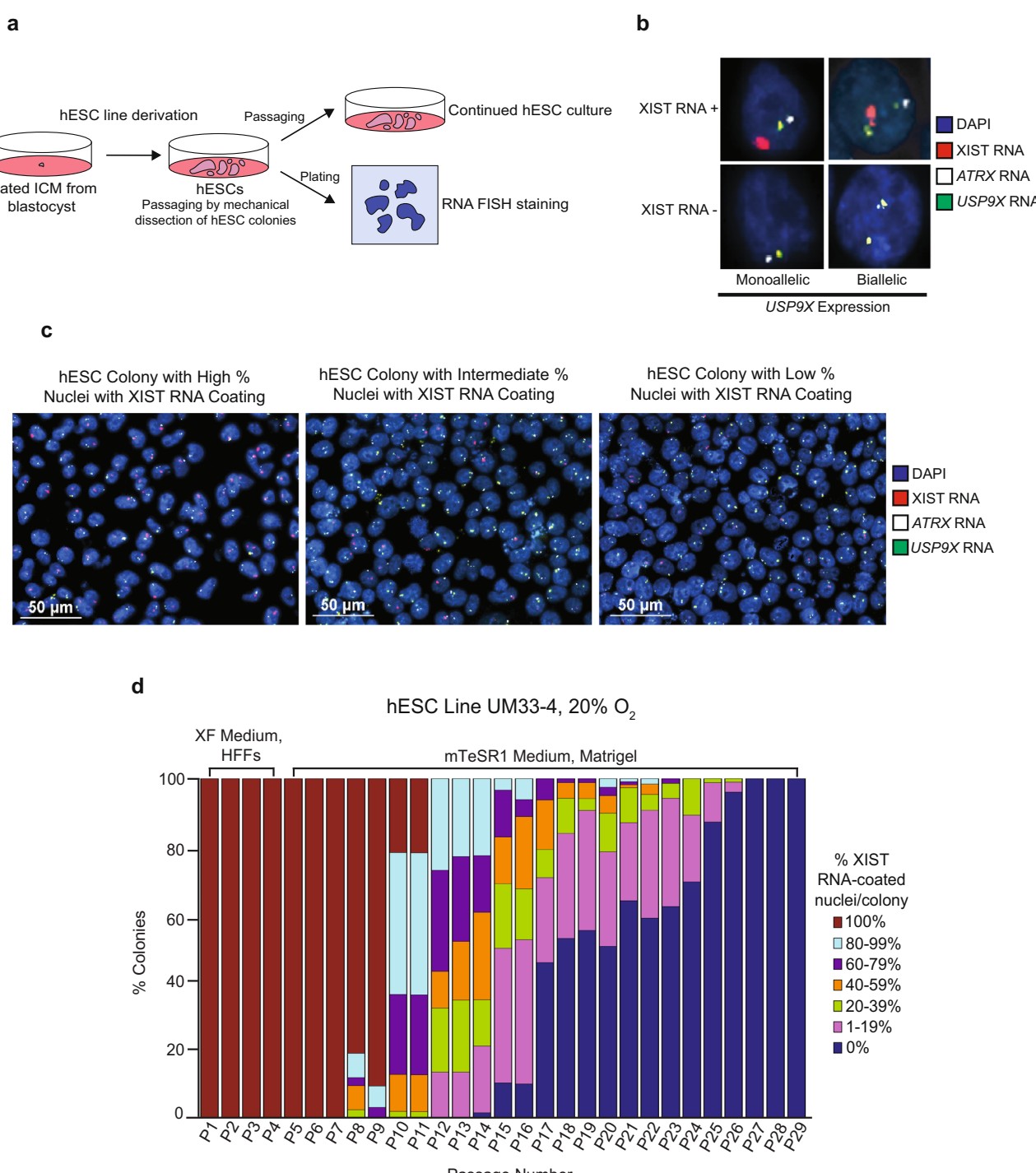

**Fig. 1 Loss of XIST RNA coating upon prolonged passaging of female hESCs. a** Schematic depicting the derivation, culture, passaging, and RNA FISH staining of hESCs in this study. **b** Representative nuclei stained to detect XIST RNA (red), RNAs from X-linked genes *ATRX* (white) and *USP9X* (green), and the nucleus with DAPI (blue). Top, representative nuclei with XIST RNA coating. Bottom, representative nuclei without XIST RNA coating. At least 100 nuclei were counted per colony for hESC RNA FISH quantification. The total number of colonies quantified at each passage range from 2–138 and are cataloged in source data. **c** Representative hESC colonies stained to detect XIST RNA (red), *ATRX* RNA (white), and *USP9X* RNA (green). Nuclei are stained blue with DAPI. At least 100 nuclei were counted per colony for hESC RNA FISH quantification. **d** Percentage of nuclei with XIST RNA coats per colony in hESC line UM33-4 derived in XenoFree (XF) medium on human fibroblast feeders (HFFs) and cultured subsequently on Matrigel under atmospheric oxygen levels (20%). The percentage of nuclei with XIST RNA coats in individual hESC colonies were stratified into 20% increments. 100% value indicates that all nuclei in a colony harbored XIST RNA coats, whereas 0% indicates that all nuclei lacked XIST RNA coats in a colony. The percentage of colonies harboring nuclei with XIST RNA coats decreased significantly with passage number (general linear model, $p = 0.002$). See also Supplementary Fig. 2. Source data are provided as a Source Data file.

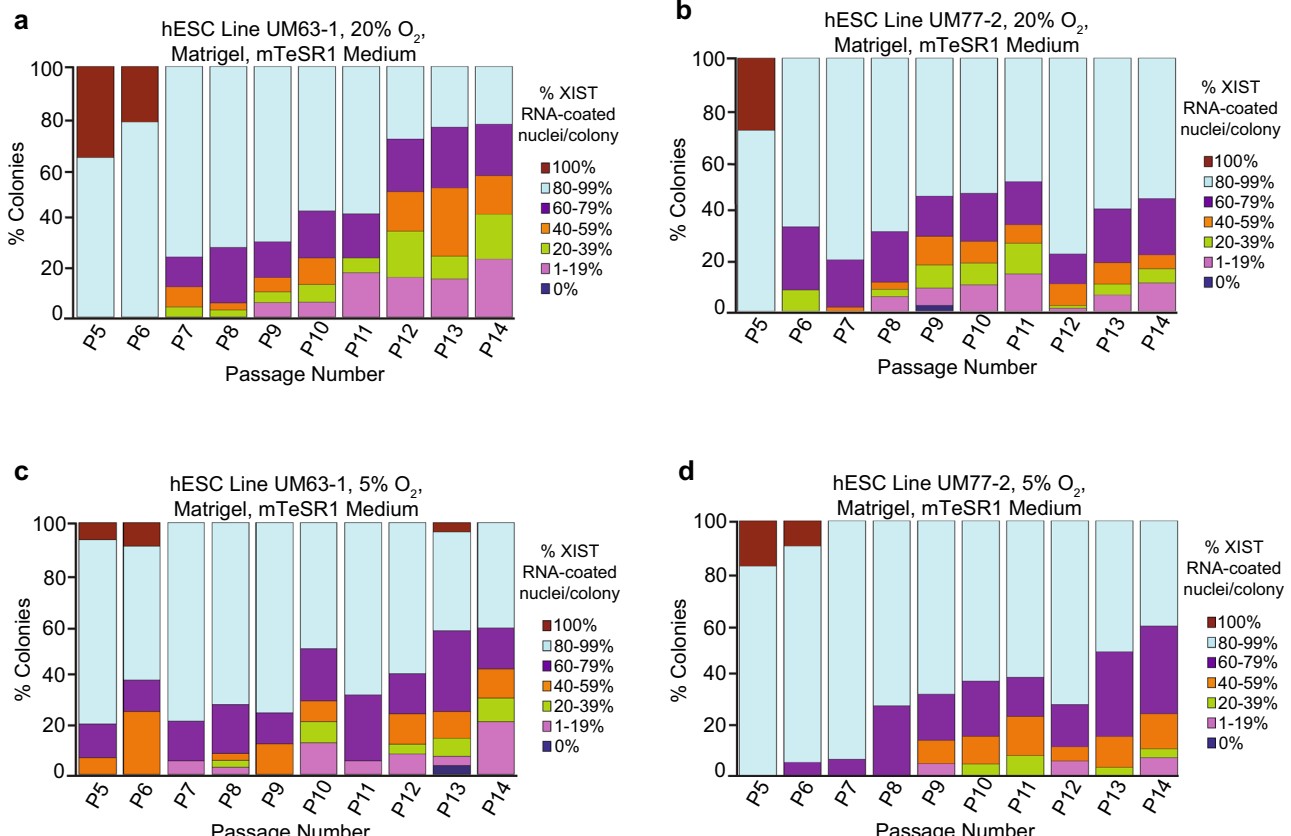

**Fig. 2 XIST RNA coating in female hESCs cultured in atmospheric vs. physiological O₂ concentration. a–d** Percentage of nuclei with XIST RNA coats per colony of hESC lines UM63-1 (**a**, **c**) and UM77-2 (**b**, **d**) cultured in parallel under 20% (**a**, **b**) and 5% (**c**, **d**) O₂ concentration on Matrigel in mTeSR1 medium. The difference in the frequency of nuclei without XIST RNA coats per colony in either cell line when cultured at physiological vs. atmospheric O₂ concentration is not significant (general linear model comparison, $p = 0.1$). See also Supplementary Fig. 2. At least 100 nuclei were counted per colony for hESC RNA FISH quantification. The total number of colonies quantified at each passage range between 11–93 and are cataloged in source data. Source data are provided as a Source Data file.

**XIST RNA coating in hESCs cultured in atmospheric vs. physiological O₂ concentration.** To investigate the underlying causes of XIST RNA coating loss in hESCs, we next cultured hESCs in atmospheric (20%) and physiological (5%) O₂ concentrations. Atmospheric O₂ has been reported to cause loss of XIST RNA expression in cultured female hESCs in some studies[34,42,47,67] but not others[29,35,43,45,68–70]. We derived two new female hESC lines on HFFs, UM63-1 and UM77-2, and passaged them from P5 to P14 onto Matrigel-coated surfaces under atmospheric and physiological O₂ concentrations in parallel. We quantified the proportion of XIST RNA-coated nuclei per colony in the UM63-1 and UM77-2 hESCs from P5 to P14 since UM33-4 hESCs in Fig. 1 lost XIST RNA coating in a significant proportion of nuclei per colony in these passages. Under atmospheric O₂ culture conditions, both the UM63-1 and UM77-2 hESC lines displayed decreasing percentages of nuclei with XIST RNA coats per colony during passaging, recapitulating the pattern observed in the UM33-4 hESC line (Fig. 2a, b; Source Data file). When cultured at physiological O₂ concentration, the UM63-1 and UM77-2 hESC lines exhibited a similar decrease in the proportion of XIST RNA-coated nuclei per colony with successive passaging (Fig. 2c, d; Source Data file). The difference in the frequency of nuclei per colony without XIST RNA coating in either cell line when cultured in physiological vs. atmospheric O₂ levels was not significant (general linear model comparison, $p = 0.1$).

Together with XIST RNA coating in the hESCs, we assayed the expression of two additional X-linked genes, *ATRX* and *USP9X*,

by RNA FISH. Under both O₂ concentrations, nuclei that had lost XIST RNAcoating maintained silencing of one *ATRX* allele but displayed an increased frequency of biallelic expression of *USP9X* across passaging (Supplementary Fig. 2; Source Data file). These results led us to conclude that culturing primed pluripotent hESCs in atmospheric vs. physiological O₂ concentration does not alter the loss of XIST RNA expression and coating in hESCs, in agreement with other reports[29,35,43,45,68–70]. Furthermore, some X-linked genes become derepressed when XIST RNA is lost in cultured hESCs, also in agreement with previously published results[31,34,35,38,42,43,58].

**Impact of hESC culture surface on XIST RNA coating and expression.** We next sought to examine other variables that could explain the loss of XIST RNA coating in cultured hESCs. After their derivation and initial culture on HFFs, the hESC lines in Figs. 1 and 2, UM33-4, UM63-1, and UM77-2, were transitioned to culture and passaging on a Matrigel-coated surface. The loss of XIST RNA coating coincided with continued passaging of the hESCs on Matrigel (Figs. 1 and 2). We thus examined whether the surface on which hESCs are cultured could contribute to X-inactivation erosion. In addition to Matrigel, hESCs can also be propagated on HFFs. We, therefore, tested whether continued culturing of hESCs on HFFs would maintain XIST RNA expression and coating. At nearly all passages, strikingly >80% of the nuclei in nearly all colonies of the UM63-1 and UM77-2 hESC lines cultured on HFFs exhibited XIST RNA coating from

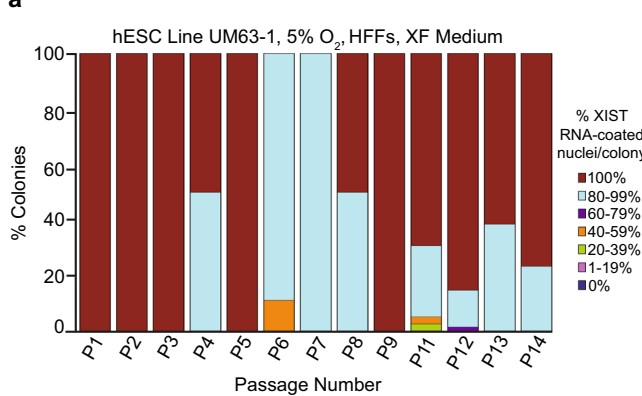

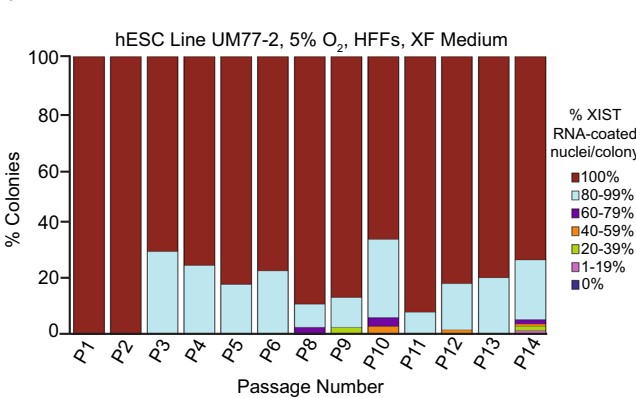

**Fig. 3 Impact of culture surface on XIST RNA coating in female hESCs.**
**a, b** Percentage of nuclei with XIST RNA coats in colonies of hESC line UM63-1 (**a**) and UM77-2 (**b**) cultured on HFFs in XF medium under 5% $O_2$ concentration. The frequency of nuclei harboring XIST RNA coats per colony in either cell line when cultured on HFFs did not decrease significantly (general linear model, $p = 0.09$). At least 100 nuclei were counted per colony for hESC RNA FISH quantification. The total number of colonies quantified at each passage range between 1 and 126 and are cataloged in source data. Source data are provided as a Source Data file.

P1-14 (Fig. 3; Source Data file). By contrast, as shown in Fig. 2, UM63-1 and UM77-2 hESCs cultured on Matrigel displayed an increasing percentage of colonies without any XIST RNA-coated nuclei through passaging (general linear model comparison, $p < 0.001$).

**Analysis of hESC culture media effect on XIST RNA coating.** hESCs grown on Matrigel vs. HFFs differ both in the culture surface as well as in the culture media used. hESCs grown on Matrigel are typically cultured in mTeSR media[62–66], whereas our hESC lines grown on HFFs were cultured with a XenoFree (XF) medium (see Methods for detailed culture media composition). We therefore investigated whether culture media could underlie the differences in XIST RNA coating observed in hESCs cultured on Matrigel vs. HFFs. We cultured hESC lines UM63-1 and UM77-2 on HFFs in mTeSR1 medium and in XF medium in parallel. Unexpectedly, we found that culturing hESCs on HFFs with XF medium stably maintained nuclei with XIST RNA coating, whereas hESCs cultured on HFFs with mTeSR1 medium displayed a significant decrease in the proportion of nuclei with XIST RNA coating per colony during equivalent passaging

(general linear model comparison, $p < 0.001$) (Fig. 4; Source Data file). We could not culture hESCs on Matrigel with XF medium, as the hESCs failed to grow and attach to the Matrigel surface under these conditions.

To further investigate the effect of culture media on XIST RNA coating in hESCs, we next asked whether XIST RNA coating patterns would change if hESCs cultured on HFFs in mTeSR1 medium were switched onto XF medium and vice versa. We, therefore, cultured on HFFs the hESC line UM63-1 initially in XF medium and then switched the culture medium to mTeSR1 medium (Supplementary Fig. 3). When cultured in the XF medium, UM63-1 displayed a small percentage of colonies without any XIST RNA-coated nuclei (Fig. 5a). This slight loss of XIST RNA coating may be due to a freeze-thaw cycle prior to culture, which has previously been linked to loss of XIST RNA coating in hESCs[34]. Whereas the two hESC lines maintained XIST RNA coating when cultured in the XF medium, both hESC lines displayed a progressive decrease in the percentage of XIST RNA-coated nuclei per colony after being switched to culturing on mTeSR1 medium (general linear model, $p = 0.01$) (Fig. 5a, b; Source Data file). Figure 5 displays the three most informative categories of percentage of XIST RNA-coated nuclei per colony (80–100%; 20–79%; 0–19%) (see Supplementary Fig. 4 and Source Data file for further stratification of the data).

When hESCs were cultured on HFFs first in mTeSR1 medium and then switched to XF medium, the frequency of XIST RNA-coated nuclei per colony stabilized (Fig. 5c, d; Supplementary Fig. 4; Source Data file). Although their proportion did not decrease significantly, XIST RNA-coated nuclei per colony also did not increase in frequency across passaging when hESCs were switched from mTeSR1 to XF medium (general linear model, $p = 0.1$), suggesting that the loss of XIST RNA expression is irreversible. From these findings, we conclude that culturing hESCs in XF medium stably maintains XIST RNA coating and that culturing hESCs in mTeSR1 medium causes an irreversible loss of XIST RNA expression.

**Lithium chloride in mTeSR1 medium as a cause of XIST RNA loss.** To determine the underlying reasons for the loss of XIST RNA coating in hESCs cultured in mTeSR1 vs. XF media, we compared the chemical composition of the two media (Methods)[64,65]. We noted that whereas mTeSR1 contains lithium chloride (LiCl), XF medium does not. Lithium is known to intersect with a number of intracellular signaling pathways[71], including, importantly, with the Wnt pathway by inhibiting GSK-3 proteins[72–74]. We thus tested if addition of LiCl to XF medium (XF with LiCl) at the concentration present in mTeSR1 (0.98 mM) would result in loss of XIST RNA coating. We derived an independent hESC line, UM90-14, and cultured the cells in parallel in XF, mTeSR1, and XF with LiCl media on HFFs under physiological oxygen concentration (5% $O_2$). As in Fig. 5, we plotted the three most informative categories of percentage of XIST RNA-coated nuclei per colony (80–100%; 20–79%; 0–19%). As expected, hESCs cultured in XF medium did not exhibit a significant decrease in the percentage of XIST RNA-coated nuclei per colony across passaging (general linear model, $p = 0.3$) (Fig. 6a; Source Data file). hESCs cultured in mTeSR1 medium displayed a significant reduction in the percentage of XIST RNA-coated nuclei per colony across passaging (general linear model, $p < 0.001$) (Fig. 6b; Source Data file). Strikingly, hESCs cultured in the XF medium with LiCl also lost XIST RNA coating during passaging in a significant percentage of nuclei per colony compared to cells cultured in XF medium (general linear model, $p < 0.001$) (Fig. 6c; Source Data file).

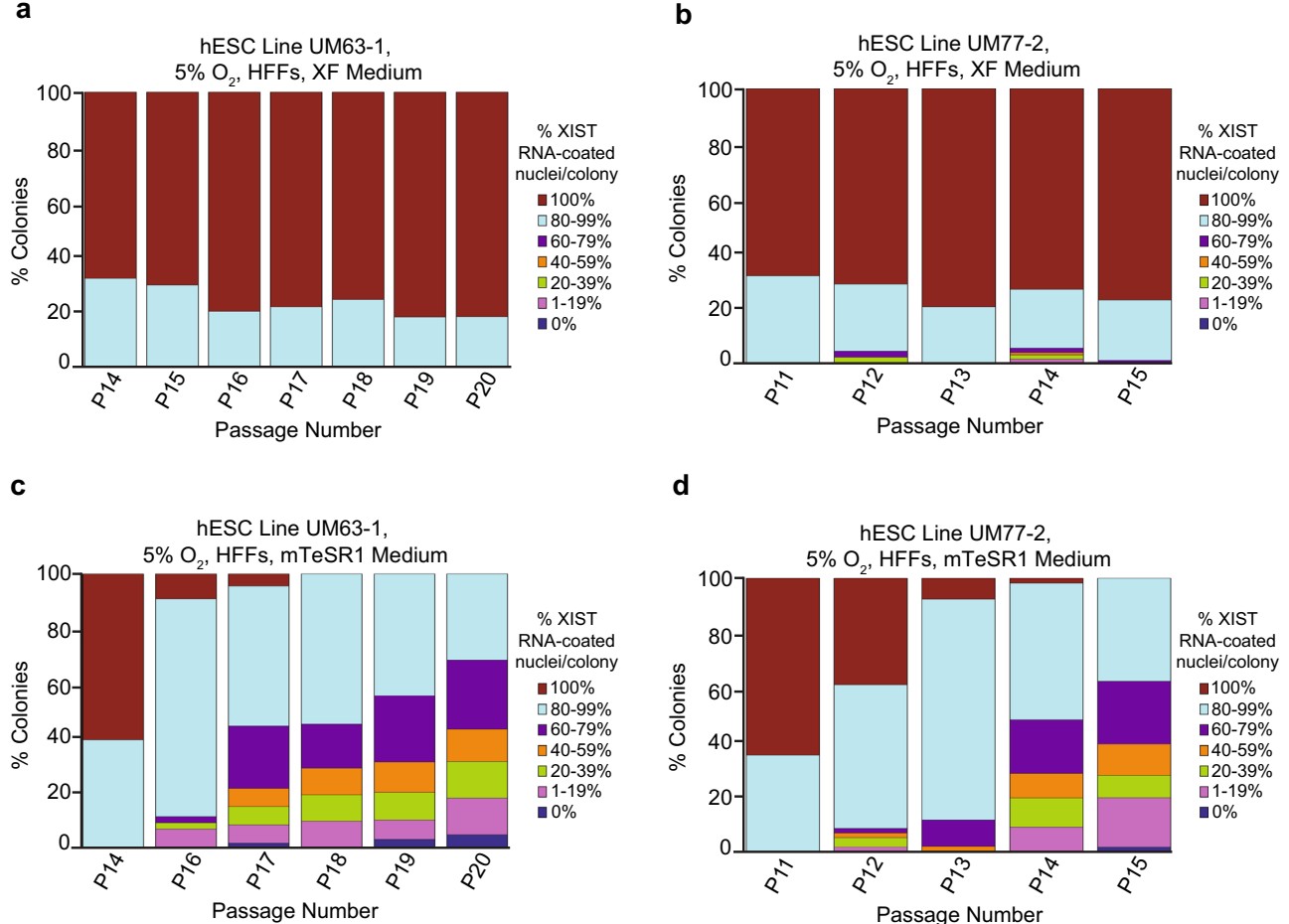

**Fig. 4 Impact of culture medium on XIST RNA coating in female hESCs. a–d** Percentage of nuclei with XIST RNA coats per colony in hESC lines UM63-1 (**a**, **c**) and UM77-2 (**b**, **d**) cultured in parallel in XF medium (**a**, **b**) and mTeSR1 medium (**c**, **d**) on HFFs. hESCs cultured with mTeSR1 medium displayed a significant decrease in nuclei with XIST RNA coating compared to hESCs cultured in XF medium during passaging (general linear model comparison; $p < 0.001$). All hESCs in this experiment were cultured in 5% $O_2$ on HFFs. The quantification data for P13-14 in **b** are taken from Fig. 3b. At least 100 nuclei were counted per colony for hESC RNA FISH quantification. The total number of colonies quantified at each passage range between 10 and 180 and are cataloged in source data. Source data are provided as a Source Data file.

**Stability of XIST RNA coating in differentiated hESCs.** The erosion of X-inactivation in cultured hESCs compelled the examination of X-inactivation stability during the differentiation of female hESCs. X-inactivation is reported to be stable upon differentiation of human PSCs in some[45] but not other studies[31,32,39,41,42,75]. We assayed XIST RNA coating in hESCs differentiated into embryoid bodies (EBs) with a commercial medium, AggreWell[TM], that is based on the mTeSR1 medium and contains LiCl[64] and with XF medium lacking the pluripotency promoting factor bFGF. bFGF is widely used as a supplement to promote growth of stem cells in vitro, including in the mTeSR1 and XF media used in this study[76]. XIST RNA coating decreased in a significant proportion of nuclei in EBs differentiated with the AggreWell[TM] medium (general linear model, $p < 0.001$) but not with the XF medium lacking bFGF (general linear model, $p = 0.3$). We next assayed XIST RNA coating in hESCs differentiated into EBs in XF medium lacking bFGF but supplemented with LiCl at the concentration present in mTeSR1 (0.98 mM) and again observed loss of XIST RNA coating in a significant proportion of nuclei (general linear model, $p < 0.01$) (Fig. 7; Source Data file). These data suggest that media containing LiCl causes loss of XIST RNA expression during the differentiation of hESCs, like during the culture of undifferentiated hESCs.

**GSK-3 inhibition and loss of XIST RNA coating in hESCs.** A prominent mode of action of lithium is inhibition of the GSK-3 pathway[71,72]. GSK-3 is a multifaceted kinase with over 100 known substrates and exists as two common isoforms, GSK-3α and GSK-3β[77–79]. GSK-3β function appears to overlap with that of GSK-3α[80], suggesting partial functional redundancy of the two proteins. A major function of GSK-3 is the negative regulation of β-catenin, which is a key mediator of the canonical Wnt signaling pathway[81]. Wnt signaling, in turn, plays an important role in hESC proliferation and cell survival[82–84]. Based on the ability of lithium to inhibit the GSK-3 proteins and the loss of XIST RNA expression in LiCl-treated cells, we focused our subsequent analyses on the effects of specific GSK-3 inhibitors on XIST RNA expression in hESCs.

We tested the effects of three GSK-3 inhibitors, Alsterpaullone, BIO, and Ly2090314, on XIST RNA coating and X-linked gene expression in hESC line UM90-14. Alsterpaullone is an ATP-competitive inhibitor of GSK-3α/β proteins[85]. BIO is a highly potent, selective, and reversible ATP-competitive inhibitor of GSK-3α/β[86]. Ly2090314 is a small-molecule inhibitor of GSK-3α/β isoforms[87]. We cultured the hESC line UM90-14 for ten passages (P6-P15) in XF medium supplemented with each of the three GSK-3 inhibitors individually. The 90-14 hESC line at P6 exhibited a slightly reduced frequency of nuclei with XIST RNA

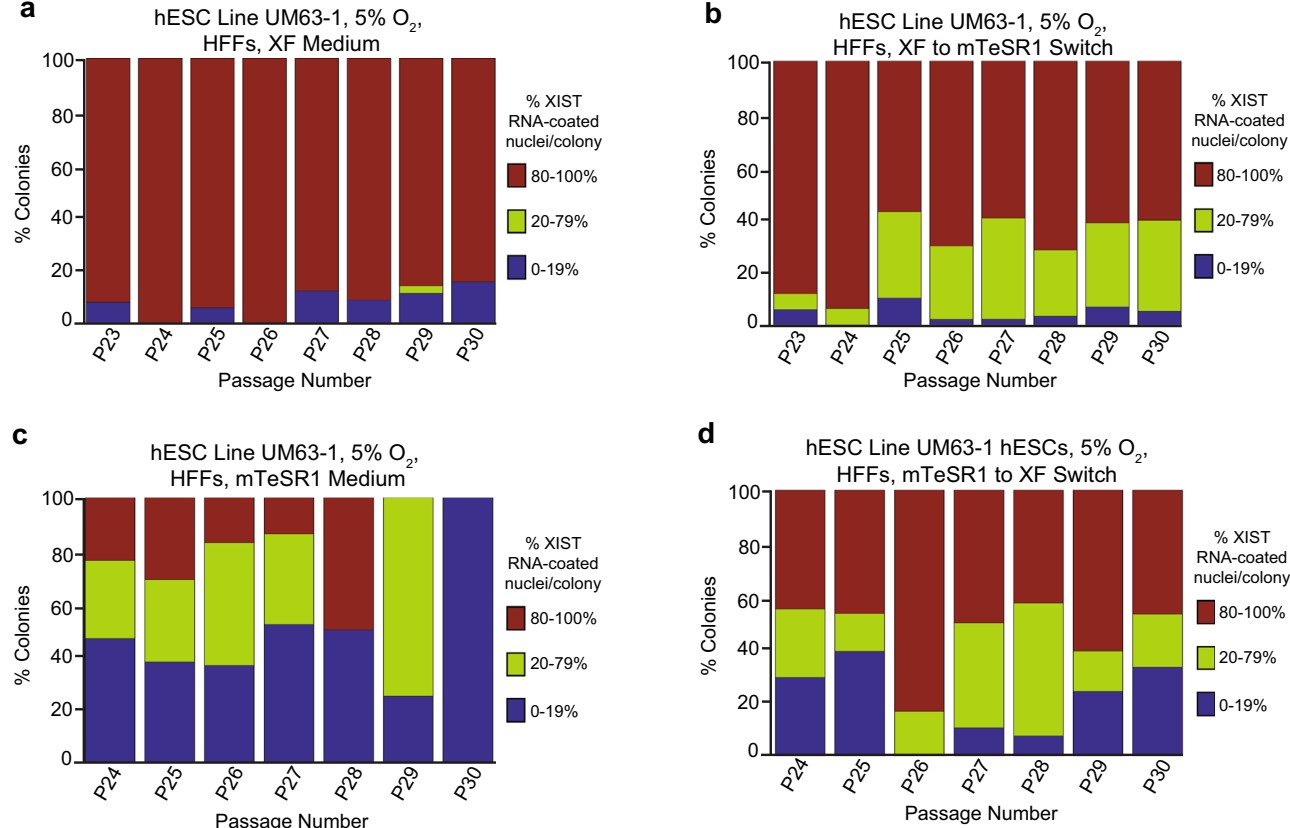

**Fig. 5 Analysis of culture media switching on XIST RNA coating in female hESCs.** Percentage of nuclei with XIST RNA coats per colony of hESC line UM63-1 (**a**) cultured continuously in XF medium and (**b**) cultured initially in XF medium and subsequently switched to mTeSR1 medium. Percentage of nuclei with XIST RNA coats per colony of hESC line UM63-1 (**c**) continuously cultured in mTeSR1 medium and (**d**) cultured initially in mTeSR1 medium and then switched to XF medium. hESCs cultured initially in XF medium and subsequently switched to mTeSR1 medium displayed a significant decrease in nuclei with XIST RNA coating during passaging compared to hESCs cultured continuously in XF medium (general linear model comparison, $p = 0.01$). hESCs cultured continuously in mTeSR1 medium displayed a significant decrease in nuclei with XIST RNA coating during passaging compared to hESCs cultured initially in mTeSR1 medium and then switched to XF medium (general linear model comparison, $p < 0.001$). All hESCs in this experiment were cultured in 5% $O_2$ on HFFs. See also Supplementary Fig. 3 and Supplementary Fig. 4. At least 100 nuclei were counted per colony for hESC RNA FISH quantification. The total number of colonies quantified at each passage range between 2 and 148 and are cataloged in source data. Source data are provided as a Source Data file.

coating compared to earlier experiments (Fig. 8; Source Data file), due possibly to the cells having been frozen and thawed prior to this series of hESC cultures[34]. The addition of each of the three GSK-3 inhibitors to the XF medium resulted in a significant reduction in nuclei with XIST RNA coating per colony upon continued passaging compared to those cultured in XF medium alone (general linear model comparison, $p < 0.001$), similar to hESCs cultured in XF with LiCl and mTeSR1 media (Fig. 8; Source Data file). A comparison of the transcriptome by RNA sequencing (RNA-Seq) of the hESCs cultured in the different media formulations above suggests that hESCs cultured in the different culture conditions above are similar but not identical to one another and to other hESCs and human epiblast cells. Furthermore, the hESCs cultured in the various culture conditions are transcriptionally distinct from human somatic cell types (Supplementary Fig. 5). Differential expression analysis of the RNA-Seq data revealed no significantly differentially expressed genes after adjusting for multiple testing, although this is likely due to sequencing only one replicate per culture condition (Source Data file).

**GSK-3 inhibition and loss of XIST RNA coating in differentiating mESCs.** We next tested if GSK-3 inhibition would lead to a loss of Xist RNA expression in differentiating female mouse (m) ESCs. Female mESCs harbor two active X chromosomes and undergo stochastic inactivation of one of the two Xs upon differentiation[88]. We differentiated the mESCs into primed pluripotent epiblast-like cells (mEpiLCs)[88] (Methods). We then cultured the mEpiLCs in the presence of the GSK-3 inhibitor CHIR99021 (CHIR) for 48 h and assessed Xist RNA coating. The mEpiLCs cultured without CHIR stably maintained Xist RNA coating, whereas mEpiLCs cultured with CHIR lost Xist RNA coating in a significant number of nuclei per colony (general linear model comparison, $p < 0.001$) (Fig. 9; Source Data file). These data suggest that the effect of GSK-3 inhibition on XIST RNA expression is conserved between hESCs and mESCs.

**GSK-3 inhibition and loss of XIST RNA coating in mEpiSCs.** We next assessed the effects of GSK-3 inhibition on Xist RNA expression in mouse epiblast stem cells (mEpiSCs). mEpiSCs are primed pluripotent stem cells that are analogous to primed pluripotent hESCs[52,54]. Like primed hESCs, mEpiSCs harbor a randomly inactivated X chromosome[14]. We found that a significant number of mEpiSCs also lost Xist RNA coating upon culture in media supplemented with CHIR (general linear model, $p < 0.001$) (Fig. 10a–c; Source Data file).

**a**

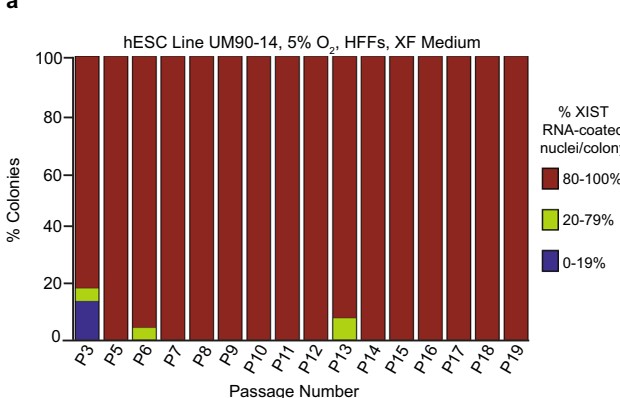

**b**

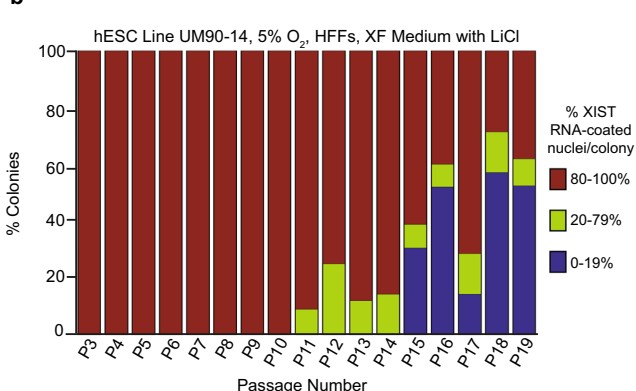

**c**

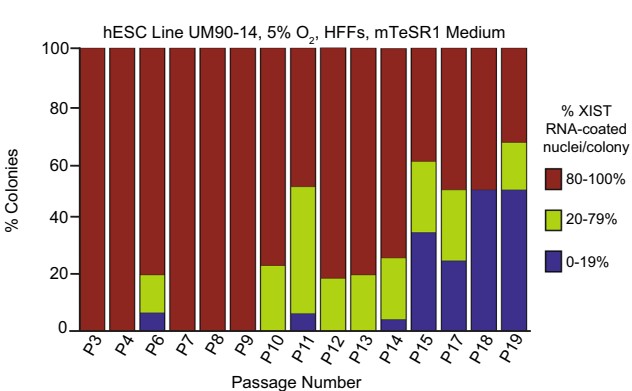

**Fig. 6 LiCl in mTeSR1 Medium as a Cause of XIST RNA Loss in Female hESCs. a–c** Percentage of nuclei with XIST RNA coating in colonies of hESC line UM90-14 cultured in XF medium (**a**), XF medium supplemented with 0.98 mM LiCl (XF with LiCl) (**b**), and mTeSR1 medium (**c**). hESCs cultured in XF medium did not display a significant decrease in nuclei with XIST RNA coats across passaging (general linear model, $p = 0.2$). hESCs cultured in XF medium with LiCl and mTeSR1 medium lost XIST RNA coating in a significant percentage of nuclei per colony during passaging compared to cells cultured in XF medium (general linear model, $p < 0.001$). All hESCs in this experiment were cultured in 5% $O_2$ on HFFs. At least 100 nuclei were counted per colony for hESC RNA FISH quantification. The total number of colonies quantified at each passage range between 5 and 78 and are cataloged in source data. Source data are provided as a Source Data file.

**Conserved TCF binding sites upstream of human and mouse XIST/Xist**. The effect of GSK-3 inhibition on XIST/Xist RNA coating in PSCs may be due directly or indirectly to the activation of Wnt signaling. Canonical Wnt signaling regulates transcription

through the binding of T-cell factor (TCF) and β-catenin to regulatory sequences of the target genes[81,89,90]. The activation of Wnt signaling is conventionally thought to only activate transcription[91], but recent studies suggest that Wnt signaling may also repress the expression of some target genes[89,90]. We therefore examined human and mouse genomic sequence 5' of the XIST/Xist transcription start sites (TSSs) and found three conserved putative TCF binding motifs within this region (Fig. 10d). We also found that sequences surrounding these motifs are also conserved between humans and mice, suggesting that these sites may serve as platforms for the binding of other transcription factors that may in turn contribute to the silencing of XIST/Xist by Wnt signaling in the two species.

## Discussion

PSCs are a model system to investigate epigenetic mechanisms that underlie cell fate transitions. X-inactivation is an experimentally tractable system to dissect epigenetic transcriptional regulation in PSCs. Much prior work has demonstrated that prolonged culture of female hPSCs results in loss of XIST RNA coating and erosion of X-inactivation[29,31,35,36,38,39,42,43,45,57,59]. Consistent with previous findings, our data rule out atmospheric $O_2$ concentration as a cause of XIST RNA loss in hESCs[35,43,45,68–70]. Instead, we found that a primary source of XIST RNA loss in hESCs is the composition of the hESC culture medium. The loss of XIST RNA expression in the hESCs is irreversible, in agreement with prior findings[35,38,39]. Of note, a recent report suggests that DNA methylation may contribute to the irreversibility of XIST RNA loss in cultured female hESCs that have undergone X-inactivation erosion[59].

Female hESCs lacking XIST RNA coating have been suggested to expand in culture due to a proliferation advantage compared to female hESCs with XIST RNA coating[29]. Our data, though, argue against this possibility because non-XIST RNA-coated hESCs did not increase in proportion when hESCs cultured in mTeSR1 medium were switched to culture with XF medium, which stably maintained but did not lead to increased frequency of nuclei with XIST RNA coating. Instead, our data suggest that LiCl in hESC culture medium can actively cause loss of XIST RNA expression. The loss of XIST RNA coating, however, did not appear to result in chromosome-wide reactivation of all silenced X-linked genes. XIST RNA loss in the hESCs, though, has been reported to result in the reactivation of a subset of silenced X-linked genes[22,38,45,49,92,93], consistent with our findings. Through increased expression of X-linked genes, the erosion of X-inactivation can negatively impact hESC physiology, for example through compromised differentiation[45].

One factor suggested to cause loss of XIST RNA expression in hESCs is the X-linked XACT RNA. XACT RNA is expressed from and coats the active X chromosome at the onset of X-inactivation[42,49,94]. *XACT* expression also precedes loss of XIST RNA coating from the inactive-X during prolonged hESC culture[42,48,95]. An analysis of published female hESC RNA-Seq data, however, suggests that *XACT* expression is likely to be transient in cultured hESCs[46,96,97].

That culture media components like LiCl can alter X-inactivation counters the conventional notion that X-inactivation is a cell autonomous process that is immune to extracellular influences. Lithium interferes with a number of cell signaling pathways, most prominently through the inhibition of GSK-3 proteins[71,72]. The inclusion of GSK-3 inhibitors in hESC culture media recapitulated the loss of XIST RNA coating observed with the addition of LiCl to the culture medium, suggesting that inhibition of GSK-3 proteins by lithium may be

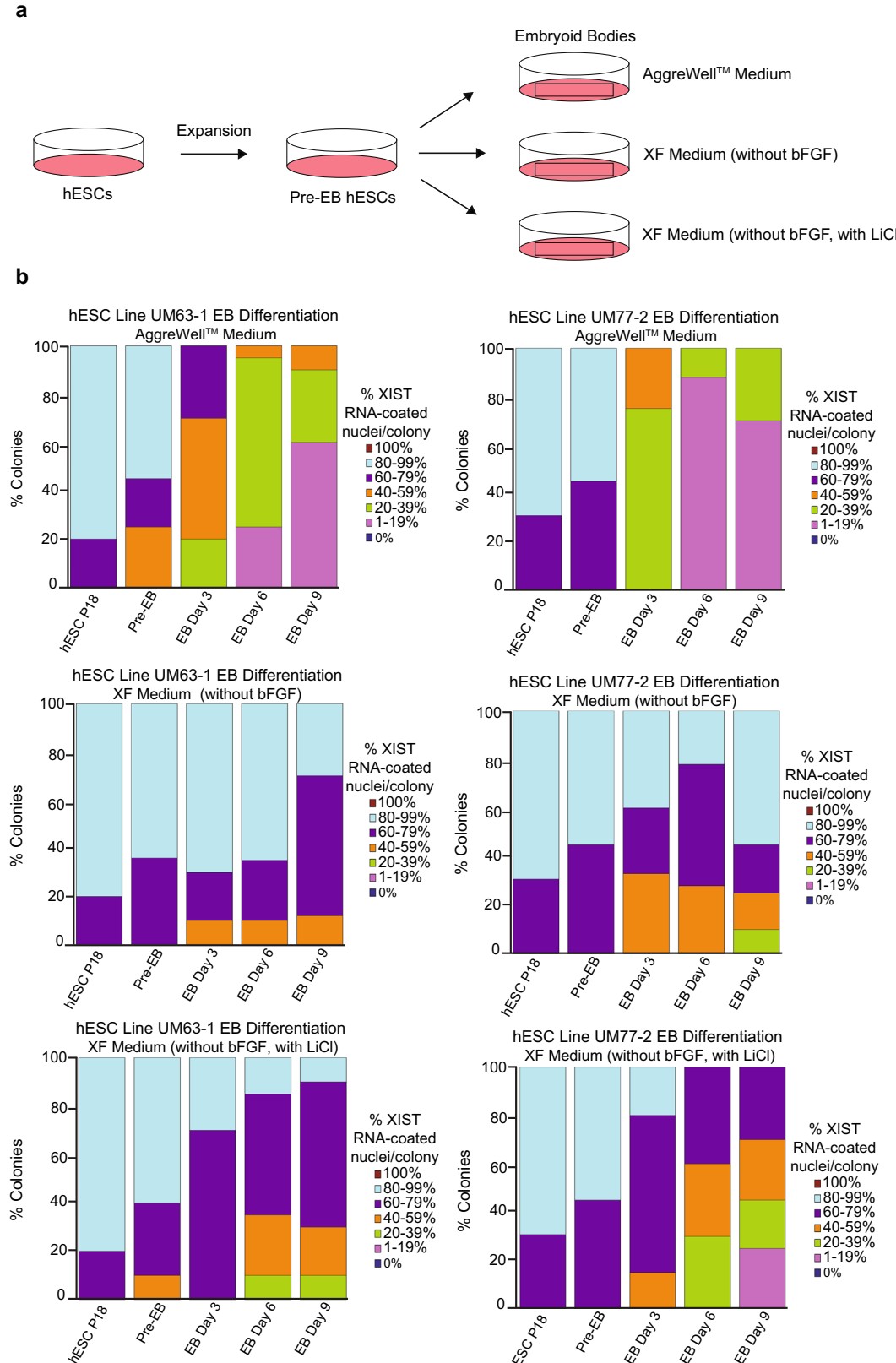

**Fig. 7 Analysis of XIST RNA coating during differentiation of female hESCs. a** Schematic of hESC differentiation into embryoid bodies (EBs) with three different media formulations: a commercially available AggreWell[TM] medium; XF medium lacking bFGF; and XF medium lacking bFGF but containing 0.98 mM LiCl. **b** Percentage of nuclei with XIST RNA coating in EBs generated from hESC lines UM77-2 and UM63-1. EBs generated and cultured in XF medium with LiCl and AggreWell[TM] medium lost a significant proportion of XIST RNA coating per colony compared to EBs generated and cultured in XF medium (general linear model comparison; $p < 0.001$). At least 100 nuclei were counted per colony for hESC RNA FISH quantification. The total number of colonies quantified at each passage range between 10 and 17 and are cataloged in source data. Source data are provided as a Source Data file.

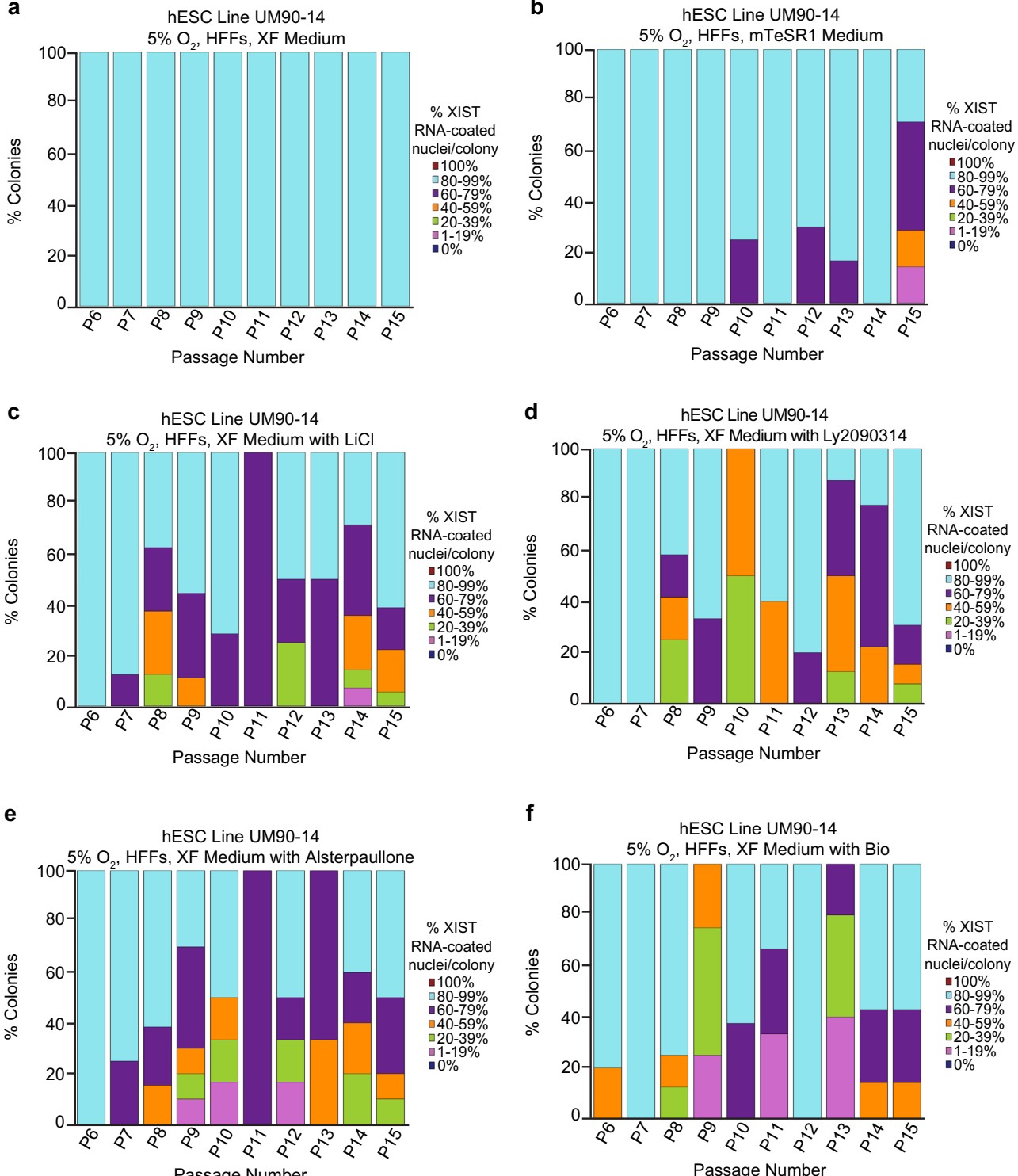

**Fig. 8 GSK-3 Inhibition and Loss of XIST RNA Coating in Female hESCs.** Percentage of nuclei with XIST RNA coating in colonies of hESC line UM90-14 cultured in XF medium (**a**); mTeSR1 medium (**b**); XF medium supplemented with 0.98 mM LiCl (XF with LiCl) (**c**); XF medium supplemented with 1.5 nM Ly2090314 (XF with Ly2090314) (**d**); XF medium supplemented with 5.0 nM Alsterpaullone (XF with Alsterpaullone) (**e**); and, XF medium supplemented with 5.0 nM BIO (XF with Bio) (**f**). hESCs cultured in mTeSR1, XF with LiCl, XF with Ly2090314, XF with Alsterpaullone, and XF with BIO media lost XIST RNA coating during passaging in a significant percentage of nuclei compared to hESCs cultured in XF medium (general linear model comparison, $p < 0.001$). All hESCs in this experiment were cultured in 5% $O_2$ on HFFs. At least 100 nuclei were counted per colony for hESC RNA FISH quantification. The total number of colonies quantified at each passage range between 1 and 18 and are cataloged in source data. Source data are provided as a Source Data file.

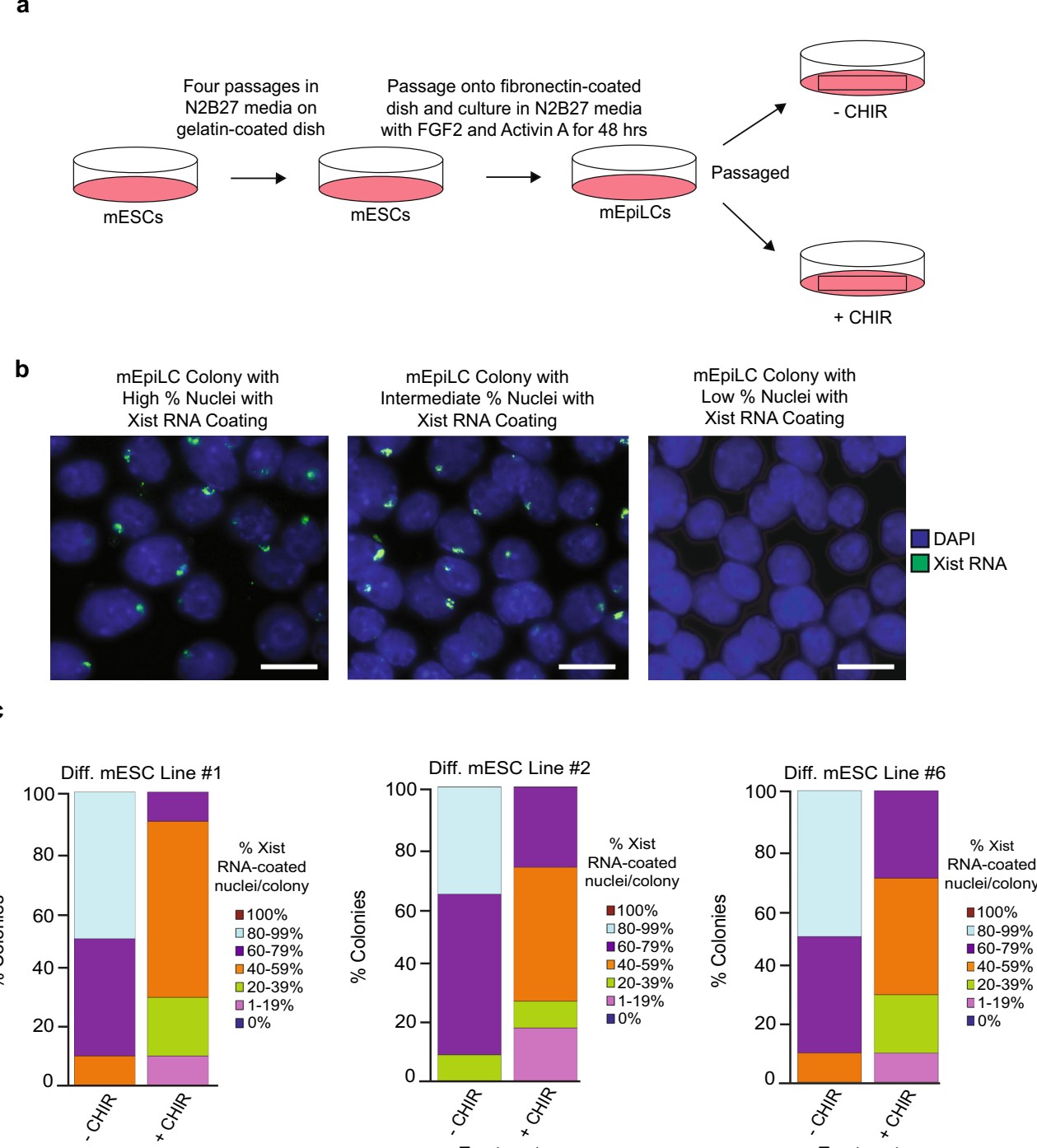

**Fig. 9 GSK-3 Inhibition and Loss of Xist RNA Coating in Differentiating Female mESCs. a** Strategy for the differentiation of mESC lines into mEpiLCs and culture of mEpiLCs with and without the GSK-3 inhibitor CHIR99021 (CHIR; 3 μM). **b** Representative images of mEpiLCs with high, intermediate, and low percent of nuclei with Xist RNA coating (green). Nuclei are stained blue with DAPI. At least ten nuclei were counted per colony in mEpiLC RNA FISH quantification. Scale bars are ~100 microns. **c** Percentage of nuclei with Xist RNA coating in differentiating mEpiLCs with and without CHIR generated from three independent ESC lines. mEpiLCs cultured with CHIR lost a significant proportion of Xist RNA coating compared to mEpiLCs cultured without CHIR in all three mEpiLC replicates (general linear model comparison, $p < 0.001$). At least ten nuclei were counted per colony for mEpiLC RNA FISH quantification. The total number of colonies quantified at each passage range between 12 and 20 and are cataloged in source data. Source data are provided as a Source Data file.

responsible for loss of XIST RNA coating in hESCs cultured in mTeSR1 medium.

A primary mode of action of GSK-3 is the negative regulation of the canonical Wnt signaling pathway[82–84]. GSK-3 phosphorylates

TCF/LEF coactivator β-catenin, which prevents β-catenin from entering nucleus[98]. GSK3 inhibition permits β-catenin to enter the nucleus, where it associates with TCF/LEF transcription factors to activate gene expression[99]. But, recent work has found that Wnt

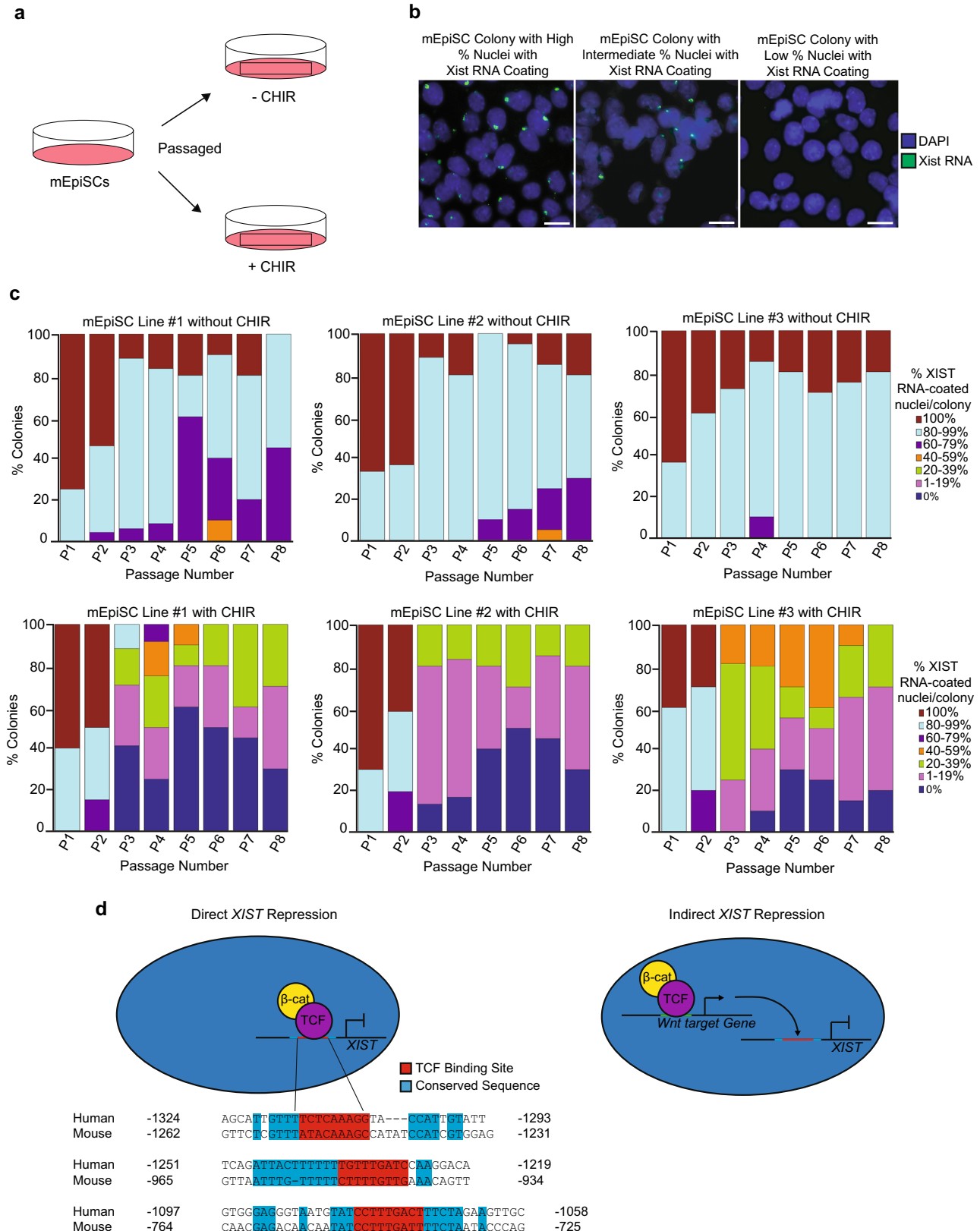

signaling can also repress gene expression in some contexts[89,90]. TCF binding at target sites together with other cofactors that silence gene expression may repress transcription of Wnt targets[89]. The putative TCF binding sites and conserved surrounding sequences upstream of *XIST* suggests that TCF together with other cofactors that may directly repress *XIST* in PSCs. It's also possible that GSK-3β and other components of the Wnt signaling pathway indirectly repress *XIST* by inducing the expression of genes that in turn silence *XIST* or by crosstalk with other signaling pathways that silence *XIST*[47,100,101]. Future work will dissect the direct vs. indirect regulation of XIST/Xist RNA expression by GSK-3 and Wnt signaling in PSCs.

**Fig. 10 GSK-3 Inhibition and Loss of Xist RNA Coating in Female mEpiSCs. a** Strategy to test the impact of GSK-3 inhibition on Xist RNA coating in mEpiSCs. **b** Representative RNA FISH images of colonies with high, intermediate, and low percentage of Xist RNA coating (green). Nuclei are stained blue with DAPI. At least ten nuclei were counted per colony in mEpiSC RNA FISH quantification. Scale bars are ~100 microns. **c** Percentage of nuclei with Xist RNA coating in three independent mEpiSC lines cultured with and without the GSK-3 inhibitor CHIR99021 (CHIR; 3 μM). All three mEpiSC lines cultured with CHIR lost a significant proportion of Xist RNA coating compared to mEpiSCs cultured without CHIR (general linear model comparison, $p < 0.001$). **d** Model of direct or indirect repression of human *XIST* and mouse *Xist* expression. Conserved mouse and human *Xist/XIST* sequences upstream of the *Xist/XIST* TSSs shown, with putative TCF binding motifs in red and surrounding conserved sequence in blue. At least ten nuclei were counted per colony for mEpiSC RNA FISH quantification. The total number of colonies quantified at each passage range between 15 and 25 and are cataloged in source data. Source data are provided as a Source Data file.

Naïve hESCs are cultured with a cocktail of inhibitors that target components of a number of cellular signaling pathways, including Wnt signaling via the inhibition of GSK-3[102–104]. GSK-3 inhibition and Wnt activation are both suggested to promote naïve hESC pluripotency[46,68,84,91,105]. In naïve hESCs, XIST RNA can be expressed from and coat both X-chromosomes[48]. However, the XIST RNA coats in naïve hESCs appear more diffuse[47], suggesting decreased *XIST* expression compared to that in primed pluripotent hESCs. This lower XIST RNA expression in naïve hESCs may potentially be due to the inclusion of GSK-3 inhibitors in naïve hESC culture media.

Although human and mouse embryos both undergo X-chromosome inactivation, the dynamics of X-inactivation differ during human and mouse embryonic development[8–12,19–22]. Furthermore, whereas human *XIST* and mouse *Xist* transcripts share some sequence homology, the genomic sequences surrounding human *XIST* and mouse *Xist* are considerably divergent[106]. The identification of conserved TCF binding motifs and surrounding sequences upstream of the human *XIST* and mouse *Xist* TSSs suggest a possible conserved role for Wnt signaling in *XIST/Xist* regulation and dosage compensation in humans and mice.

Our findings suggest a careful assessment and selection of culture media in order to faithfully maintain the epigenetic state of hESCs. Given that lithium chloride and other GSK-3 inhibitors can cause loss of XIST RNA expression, the use of media without LiCl and other GSK-3 inhibitors may be more appropriate in the maintenance and differentiation of primed pluripotent hESCs. The inclusion of GSK-3 inhibitors in naïve hESC culture media and their impact on the cellular epigenome may also require careful investigation.

Although our study has shown that the inclusion of LiCl and GSK-3 inhibitors in culture media impedes expression of XIST RNA in PSCs, it does not exclude other causes of loss of XIST RNA expression in the PSCs. In addition to GSK-3 inhibition, sources of cellular stress, such as freeze-thaw cycles of hESCs, may also alter XIST RNA expression[34].

## Methods

**Ethics**. Written informed consent for human embryo donation was obtained from both gamete providers as outlined by NIH guidelines, and hESC line derivation was performed under University of Michigan's Institutional Review Board approved study, "Derivation of human Embryonic Stem Cells" (IRB-Med; HUM00028742). This study was also approved by the Human Pluripotent Stem Cell Research Oversight (HPSCRO) Committee (HPSCRO record #1035).

**Antibodies, plasmids, and reagents**. Antibodies were used at the following concentrations and obtained from the following sources: Mouse anti-TRA 1-60, pluripotency marker, 1:200 dilution, Millipore #MAB4360; Goat anti-OCT3/4, pluripotency marker, 1:300 dilution, Santa Cruz Biotechnology #sc-8628; Rabbit anti-SOX2, pluripotency marker, 1:800 dilution, EMD Millipore #AB5603; Rabbit anti-NANOG, pluripotency marker, 1:150 dilution, Abcam #ab21624; Mouse anti-SSEA4, pluripotency marker, 1:100 dilution, Millipore #MAB4304; Donkey anti-Rabbit, secondary antibody, NANOG 1:200 dilution, SOX2 1:800 dilution, Jackson ImmunoResearch #711-165-152; Donkey anti-Goat, secondary antibody; 1:800 dilution, Jackson ImmunoResearch #705-096-147; Donkey anti-Mouse, secondary antibody; TRA 1-60: 1:100 dilution; SSEA4: 1:800 dilution, Jackson ImmunoResearch #715-165-150. *XIST* BAC (RP13-183A17), *ATRX* BAC (RP11-42M11), and

*XIST* fosmid (G135P63425C4) were obtained from the BAC PAC Resources Center (https://bacpacresources.org). *USP9X* BAC was obtained from Invitrogen (http://clones.thermofisher.com/bacpacsearch.php; BAC Human CTD 3174G14). Normal Goat Serum (#31872) was obtained from Invitrogen. Venor™ GeM Mycoplasma Detection Kit was obtained from Sigma-Aldrich (#MP0025); RNeasy Kit was obtained from Qiagen (#74104); High Capacity cDNA Reverse Transcription Kit was obtained from Applied Biosystems (#4368814); DAPI was obtained from Invitrogen (#D21490); Cy3-dCTP and Cy5-dCTP were obtained from GE Healthcare (#PA53021 and #25801087, respectively); Fluorescein-12-UTP was obtained from Roche (#11427857910); NaCl was obtained from ThermoFisher Scientific (#BP358); Sucrose was obtained from ThermoFisher Scientific (#BP220); MgCl$_2$ was obtained from ThermoFisher Scientific (#AA12315A7); PIPES was obtained from Sigma-Aldrich (#P6757); Paraformaldehyde was obtained from Electron Microscopy Sciences (#15710); Vectashield was obtained from Vector Labs (#H-1000); Deionized formamide was obtained from ISC Bioexpress (VWR:#0606); SSC was obtained from Ambion (#AM9765); BSA was obtained from New England Biolabs (NEB:#B9001S); Yeast tRNA was obtained from Invitrogen (#15401–029); Sheared boiled salmon sperm was obtained from Invitrogen (#15632–011); Dextran sulfate was obtained from Millipore (#S4030); Blue Hoechst dye was obtained from ThermoFisher Scientific (#33258); Matrigel was obtained from Corning (#354277); mTeSR1 medium was obtained from StemCell Technologies (#85850); Knockout™ DMEM was obtained from Gibco (#10829); Xenofree Knockout™ Serum Replacement was obtained from Gibco (#12618012); Glutamax™ was obtained from Gibco (#35050–061); β-mercaptoethanol was obtained from Sigma-Aldrich (#M6250, #M7522); nonessential Amino Acid 100× was obtained from Gibco (#11140-05); Collagenase Type IV (#17104-019) was obtained from Gibco; Basic Fibroblast Growth Factor (bFGF) for culture of hESCs was obtained from MilliporeSigma (#GF003AF-100UG); DMEM F12 was obtained from Gibco (#11330); LY2090314 was obtained from Sigma-Aldrich (SML1438); BIO was obtained from Sigma-Aldrich (#B1686); Alsterpaullone was obtained from Sigma-Aldrich (#126870); AggreWell™ medium was obtained from StemCell Technologies (#05893); bFGF (#233-FB) for differentiation of mESCs was obtained from R&D Systems; L7 passage solution was obtained from Lonza, (#FP-5013); G2™ Plus medium (#10132), Ovoil™ (#10029), and G-MOPS™ plus medium (#10130) were obtained from Vitrolife; Neurobasal medium (#21103049) was obtained from ThermoFisher Scientific; L-glutamine (#25030) was obtained from Gibco; Penicillin–streptomycin (#15070–063) was obtained from Gibco; B27 supplement (#17504) was obtained from Invitrogen; N2 supplement (#17502048) was obtained from Invitrogen; CHIR99021 (#04–0004) was obtained from Stemgent; MEK inhibitor PD0325901 (#04–0006) was obtained from Stemgent; LIF was obtained (#ESG1106) was obtained from Millipore; Activin A (#338-AC) was obtained from R&D Systems; Fibronectin (#F1141) was obtained from Sigma; bFGF (#233-FB) was obtained from R&D Systems; SsoAdvance SYBR Green Supermix was obtained from Bio-Rad (#1725272); 3 M Sodium Acetate was obtained from Teknova (#S0298); Triton X-100 was obtained from Fisher Scientific (#EP151); TRIzol was obtained from Life Technologies (#15596-018); 60 mm tissue culture dishes were obtained from BD Falcon (#353652). 1X PBS (#10010023) was obtained from ThermoFisher Scientific.

**hESC derivation, expansion, and characterization**. The sources and identifying information for all hESC and feeder cell lines used in this study are included in Supplementary Table 1. All hESC lines used in this study were derived from human embryos generated for infertility treatments and were donated to the University of Michigan for one of two reasons: (1) they were no longer needed for reproductive purposes by the donating couple; or (2) they were considered unsuitable for implantation following preimplantation genetic testing (PGT). Voluntary IRB-approved informed consent was obtained from each gamete provider of the embryos donated for hESC derivation. Each donor was informed, in writing and verbally, that donated embryos would be used for attempted derivation of hESCs. Each donor couple was informed of other disposition options, and that neither consenting nor refusing to donate embryos for research would affect the quality of care provided to the potential donors. There was a clear separation between the prospective donors' decision to create human embryos for reproductive purposes and the prospective donors' decision to donate human embryos for research purposes. No payments, cash or in kind, were offered for the donated embryos. Finally, all human embryo donations were documented with evidence of

compliance with each of the fifteen (15) elements of Section II (A) of NIH Guidelines, submitted to, reviewed, and accepted on the NIH hESC registry (https://grants.nih.gov/stem_cells/registry/current.htm).

Vitrified human embryos were warmed according to the donating fertility healthcare provider vitrification/warming protocol specific for the donated embryos. Warmed day (d) 5 embryos were cultured in equilibrated 50 ul drops of G2 Plus medium (Vitrolife, #10132), overlaid with Ovoil (Vitrolife, #10028) in a 37 °C humidified incubator at 6% $CO_2$/5% $O_2$/89% $N_2$ (low $O_2$) until blastocysts were fully expanded or had completed the hatching process from the zona pellucida. The inner cell mass (ICM) of each embryo was either removed immediately or the embryo was incubated for several hrs prior to removal of the ICM, depending on the degree of blastocoele expansion. Blastocyst morphology was scored according to the Gardner and Schoolcraft criteria[107].

The ICM of each expanded blastocyst was isolated via laser-assisted microdissection (Hamilton Thorne Biosciences) in 10 ul drops G-MOPS Plus medium (Vitrolife, #10130), overlaid with Ovoil (Supplementary Fig. 1a). The blastocysts were immobilized with a holding pipette (20 μm I.D., 40 μm O.D.). Once adequate tension was established, a calibrated visual target was employed to align the region of the blastocyst to be breached. Several pulses of infrared laser (300 mV, 0.2 ms) were delivered directly to the hatched embryo or through the zona pellucida opposite the location of the ICM, until the trophectoderm (TE) was exposed. The number of laser pulses necessary to microdissect the ICM varied for each individual blastocyst.

The blastocysts with breached zona pellucida were repositioned to a holding pipet on the opposite side (30 μm I.D., 60 μm O.D.), and were immobilized with adequate tension on the exposed, laser-breached area of the TE. On the opposite side, adequate tension was applied to the zona pellucida with the smaller (20 μm I.D., 40 μm O.D.) holding pipet and the TE/ICM complex was extracted from the zona pellucida with gentle pulling motion of the holding pipets. Each ICM was then microdissected from the overlying TE cells with infrared laser pulses, ensuring that the distance of the laser pulses to the ICM was adequate (300 mW 0.2 ms, Hamilton Thorne Biosciences).

The dissected ICM were plated on neonatal HFFs (Global Stem, #GSC-3002), inactivated by g-irradiation at $3.7 \times 104$ cells/cm$^2$, in a 35 mm tissue culture dish on a coverslip (Falcon, BD) pre-coated with 0.1% recombinant human gelatin (Fibrogen) (Supplementary Fig. 1a; considered passage 0, or P0). The plated ICM was cultured in hESC XenoFree (XF) culture medium [Knockout DMEM (Gibco, #10829)] containing 20% XF Knockout Serum Replacement (KSR) (Gibco, #12618012), 1 mM Glutamax$^{TM}$ (Gibco, #35050–061), 0.1 mM β-mercaptoethanol (Sigma-Aldrich, #M6250), 10 mM nonessential Amino Acid 100x (Gibco, #11140-05), 4 ng/ml animal-free basic human recombinant Fibroblast Growth Factor (bFGF) (MilliporeSigma, #GF003AF-100UG). The ICM and cell outgrowths were cultured under 5% $CO_2$/5% $O_2$/90% $N_2$ (low $O_2$) in a humidified incubator at 37 °C. Attachment of ICM was evaluated 48 h after plating onto HFFs and the hESCs were first passaged (P1) 4–6 days after initial evaluation (Supplementary Fig. 1a). Following manually/mechanically splitting the initial hESC colony, portions were transferred to new HFFs with the above culture medium or as specified for individual experiments.

The culture conditions for cell expansion during P1-3 on HFFs were 5% $CO_2$/5% $O_2$/90% $N_2$ (low $O_2$) at 37 °C. During this early hESC derivation passages, epiblast-like structures were identified, grown, split, and expanded until well-recognized hESC colonies were observed [P2-3 (~d14-21); Supplementary Fig. 1a]. Passaging was performed without enzymatic treatment and with manual and mechanical cutting of hESC colonies with glass microtools into ~100 μM pieces. At an early passage, hESCs were confirmed to be female (XX chromosome complement) by comparative genomic hybridization array (aCGH) following whole genome amplification (Supplementary Fig. 1d; See below). Specific experimental culture condition decision branching was performed at P1-3 with plating of hESCs at each indicated passage for analysis of XIST RNA coating and X-linked gene expression. If hESCs continued to be passaged on HFFs, culture conditions were as indicated in the Results section with low $O_2$ or 5% $CO_2$/20% $O_2$/remainder air (high $O_2$) at 37 °C. In some experiments, as described in the Results section, hESCs were transferred onto feeder-free matrix, Matrigel (Corning, #354277), and cultured either with the mTeSR1 medium[64,65] (StemCell Technologies, #85850) together with 20% KSR or continued with the XF medium. A full list of the contents of mTeSR1 medium can be found in ref. [64]. Early passage (P4-10) cryopreservation of lines was performed using vitrification[108]. Following experiments comparing XF culture medium and mTeSR1 medium, the composition of the two culture media identified the presence of lithium chloride (LiCl; 0.98 mM) in mTeSR1 and its absence in the XF culture medium. To test the impact of LiCl, 0.98 mM LiCl was added XF culture medium. The experiments outlined in Fig. 8 were performed with addition of GSK-3 inhibitors to XF medium at the following reported optimized concentrations: LY2090314, 1.5 nM; BIO, 5.0 nM; Alsterpaullone, 4.0 nM[85–87]. The 90-14 hESC line was vitrified and warmed prior to culturing for this experiment.

All four hESC lines used in this study (UM33-4, UM63-1, UM77-2, and UM90-14) were characterized for: (1) absence of mycoplasma; (2) presence of pluripotency markers (Supplementary Fig. 1b, c); (3) normal XX female karyotype (Supplementary Fig. 1e; see below); (4) short-tandem repeat (STR) analysis (see below); and, (5) embryoid body formation and presence of lineage markers for endoderm, mesoderm, and ectoderm (Supplementary Fig. 1f).

**Testing hESCs for mycoplasma contamination**. All four hESC lines used in the study were tested for mycoplasma contamination using a PCR-based assay (Venor™ GeM Mycoplasma Detection Kit, Sigma, #MP0025) and gel electrophoresis. The assay contained a PCR-negative control (no polymerase), a positive control (non-infectious DNA fragments of *Mycoplasma orale* genome, band size 267 bp), and an internal control (internal sequence of HTLV-I tax gene, presence of amplification band at 191 bp and no band at 267 bp).

**Immunocytochemistry**. All four hESC lines were cultured on coverslips prior to immunocytochemistry. Culture media was aspirated and cells were washed with 1X PBS (Fisher Scientific, #10010023) and fixed in a solution containing 4% paraformaldehyde (Electron Microscopy Sciences, #15710) and 4% Sucrose (Thermo-Fisher Scientific, #BP220) for 15 min at room temperature. The cells were then permeabilized with 0.1% Triton X-100 (Fisher Scientific, #EP151) for 5 min at room temperature and were blocked in a 5% Normal Goat Serum (Invitrogen # 31872) solution for 1 h at room temperature. The cells were stained with primary antibodies overnight at 4 °C. Immunocytochemical analysis was performed on all hESC lines with specific antibodies against pluripotency markers SOX2 (rabbit; EMD Millipore #AB5603; 1:800 dilution); NANOG (rabbit; Abcam #ab21624; 1:150 dilution); OCT3/4 (goat; Santa Cruz Biotechnology #sc-8628; 1:300 dilution); TRA1-60 (mouse; Millipore #MAB4360; 1:200 dilution); and, SSSEA-4 (mouse; Millipore #MAB4304; 1:100 dilution). Primary antibody staining was followed by three washes in 1× PBS (Fisher Scientific, #10010023) for 5 min each. After washing, samples were incubated with fluorescent secondary antibodies: donkey anti-rabbit secondary antibody coupled to the Cy3 fluorophore (ImmunoResearch #711-165-152) was used at 1:200 dilution with the rabbit anti-NANOG primary antibody and at 1:800 dilution with the rabbit anti-SOX2 primary antibody; donkey anti-goat secondary antibody coupled to the FITC fluorophore (Jackson ImmunoResearch #705-096-147) was used at 1:800 dilution with the goat anti-OCT3/4 primary antibody; donkey anti-mouse secondary antibody coupled to the Cy3 fluorophore (Jackson ImmunoResearch #715-165-150) was used at the following dilutions with the primary antibodies: mouse anti-TRA 1-60, 1:100 dilution; and, mouse anti-SSEA4, 1:800 dilution. Nuclei were labeled with Hoechst co-staining (blue; Hoechst ThermoFisher Scientific #33258) and imaged using an Olympus IX71 microscope. Colony morphology was simultaneously assessed by phase-contrast brightfield microscopy.

**Quantitative real-time PCR for pluripotency and cell lineage markers**. RNA was isolated from all four hESC lines using an RNA isolation kit (QIAGEN RNeasy Kit, #74104) and reverse transcribed into cDNA using a High Capacity cDNA Reverse Transcription Kit (Applied Biosystems, #4368814). Expression of pluripotency markers, *OCT3/4*, *NANOG*, and *SOX2*, and tissue-specific markers, Alpha-fetoprotein (*AFP*) and GATA-4 (endoderm), Brachyury (*BRACHY*) and vascular endothelial cadherin (*VE-CAD*; medoderm), and neuron-specific class III β-tubulin (*TUJ-1*) and type I intermediate filament chain keratin 18 (*KRT-18*; ectoderm) were measured in triplicate by quantitative real-time PCR using primer sets listed in Supplementary Table 2. Relative fold expression for genes of interest were calculated using the comparative CT method with β-ACTIN (*ACTB*) as the internal control. Expression levels of pluripotency markers in the hESCs were normalized to *ACTB* expression.

**Karyotyping**. G-banding was performed on 20 metaphase spreads of all four UM hESCs at passages ranging from P6-31 (Cell Line Genetics, Madison, WI). Meta-phase spreads were evaluated at 100X with a Leica GSL Scanner (100X objective, Leica GSL 120 CytoVision (Leica MicroSystems, Buffalo Grove, IL)) with band–count resolution of ~475. Cytogenic analysis demonstrated all hESC lines analyzed were female 46XX, as shown in Supplementary Table 3.

**Short-tandem repeat analysis**. Cell Line DNA fingerprinting was performed on all four hESC lines used in the study (Cell Line Genetics, Madison, Wisconsin). Analysis of fifteen short-tandem repeat (STR) loci, plus the gender determining locus, *Amelogenin*, confirmed the presence of a single human cell line that is unique from lines published in the ATCC, NIH, or DSMZ websites.

**Characterizing differentiation potential of hESCs**. hESCs were cultured on Matrigel (Corning, #354277) in mTeSR1 medium (StemCell Technologies, #85850) as described above with the following culture conditions: 37 °C, 5% $CO_2$/20% $O_2$/remainder air until reaching ~80% confluency and harvested mechanically with scrapers. Detached cells were transferred and cultured in a 60 mm petri dish (BD Falcon, #353652) containing 6 ml of AggreWell$^{TM}$ medium for embryoid body (EB) differentiation (StemCell Technologies, #05893) with media changes every other day for 14d. After 8d of differentiation, EBs were collected and RNA was extracted using QIAGEN RNeasy Kit (#74104) from all four hESC lines. The reverse transcription (RT) of total RNA to single-stranded cDNA was performed with the High Capacity cDNA Reverse Transcription Kit (Applied Biosystems, #4368814). Cell differentiation was assessed by profiling expression of molecular markers associated with the three somatic germ layers using real time reverse transcription PCR using primer sets listed under Quantitative Real-Time PCR for

Pluripotency and Lineage Markers (Bio-Rad SsoAdvance SYBR Green Supermix Bio-Rad, #1725272).

**hESC differentiation into embryoid bodies for XIST RNA coating analyses**. hESC lines UM77-2 and UM63-1 were grown, maintained, and expanded on HFFs in XF medium as described above. Colonies were cut, detached, and transferred to a 60 mm culture dish (BD Falcon, #353652) and subsequently cultured with one of three designated EB culture media: (1) XF medium without bFGF; (2) XF medium without bFGF and with 0.98 mM LiCl; and, (3) Aggrewell™ medium (StemCell Technologies, #05893) (mTeSR1-based medium containing LiCl). EBs were cultured for 9 days at 37 °C, 5% $CO_2$ and 20% $O_2$. EBs were dissociated with L7 passage solution (Lonza, #FP-5013) prior to collection at days 3, 6, and 9. Dissociated EBs were plated on Matrigel-coated (Corning, #354277) coverslips in each of the above media formulations for 48 h before processing for RNA FISH.

**mEpiLC generation and culture**. mESC lines were derived from individual E3.5 preimplantation mouse embryos, which were plated on quiescent mouse embryonic fibroblast (MEF) feeder cells in ESC derivation media consisting of Knockout DMEM (Gibco, #10829–018), Knockout serum replacement (Invitrogen, #10828–028), l-glutamine (Gibco, #25030), MEM nonessentials amino acids (Gibco, #11140–050), β-Mercaptoethanol (Sigma, #M7522), Penicillin–streptomycin (100×) (Gibco, #15070–063), GSK3 inhibitor CHIR99021 (Stemgent, #04–0004), MEK inhibitor PD0325901 (Stemgent, #04–0006) and LIF ($10^7$/mL) (Millipore, #ESG1106). Plates were incubated at 37 °C, 5% $CO_2$ for 48 h. On day 3 post-plating, mESC derivation media was replaced with fresh mESC derivation media. Outgrowths became prominent at 4–5 days post-plating, and were dissociated in 0.05% trypsin (Invitrogen, #25300-054). Dissociated embryos were plated individually into wells of a MEF-plated 96-well plate with ESC derivation media. mESC colonies became evident over the next 2–3 days and were maintained in mESC culture media consisting of Knockout DMEM (Gibco, #10829–018), fetal bovine serum embryonic stem cell qualified (ES-FBS) (Bio-Techne, #S10250), Knockout serum replacement (Invitrogen, #10828–028), l-glutamine (Gibco, #25030), MEM nonessential amino acids (Gibco, #11140–050), β-Mercaptoethanol (Sigma, #M7522), GSK3 inhibitor CHIR99021 (Stemgent, #04–0004), MEK inhibitor PD0325901 (Stemgent, #04–0006) and LIF ($10^7$/mL) (Millipore, #ESG1106).

To generate mEpiLCs, mESC lines were passaged into 2i culture conditions [N2B27 medium consisting 50% DMEM/F12 (ThermoFisher Scientific, # 11320033), 50% neurobasal medium (ThermoFisher Scientific, #21103049), 2 mM L-glutamine (GIBCO, #25030), 0.1 mM β-mercaptoethanol (Sigma,#M7522), N2 supplement (Invitrogen #17502048), B27 supplement (Invitrogen #17504-044), supplemented with 3 μM GSK-3 inhibitor CHIR99021 (CHIR) (Stemgent #04–0004), 1 μM MEK inhibitor PD0325901 (Stemgent #04–0006), and 1000 U/ml LIF (Millipore #ESG1106)] at 5% $CO_2$ and grown in gelatin-coated tissue culture dishes for 4 passages[109,110].

To differentiate mESCs into EpiLCs, mESCs were cultured in N2B27 medium supplemented with 10 ng/ml bFGF (R&D Systems, #233-FB) and 20 ng/ml Activin A (R&D Systems, #338-AC) in Fibronectin (15 μg/ml) (Sigma #F1141) coated tissue culture dishes for 48 h. The mEpiLCs were then cultured without or with 3 μM CHIR99021 (CHIR) (Stemgent #04–0004) for an additional 48 h in N2B27 medium. A concentration of 3 μM was selected for CHIR because this is the CHIR concentration used in naïve mESC culture[88]. mEpiLCs utilized for RNA FISH staining were cultured, permeabilized, and fixed on fibronectin-coated (15 μg/ml) (Sigma #F1141) coverslips.

**mEpiSC derivation and culture**. mEpiSCs were derived from individual E3.5 preimplantation mouse embryos, which were plated on quiescent mouse embryonic fibroblast (MEF) feeder cells in K15F5 medium containing Knockout DMEM (GIBCO, #10829–018) supplemented with 15% Knockout Serum Replacement (Gibco, #A1099201), 5% ES-FBS (GIBCO, #104390924), 2 mM L-glutamine (Gibco, #25030), 1X nonessential amino acids (Gibco, #11140–050), and 0.1 mM β-mercaptoethanol (Sigma, #M7522). After 5–6 days, blastocyst outgrowths were dissociated partially with 0.05% trypsin (Invitrogen, #25300-054). The partial dissociates were plated individually into a 1.9-cm² well containing a MEF feeder layer and cultured for an additional 4–6 days in K15F5 medium. The culture was then passaged by a brief exposure (2–3 min) to 0.05% trypsin/EDTA with gentle pipetting to prevent complete single-cell dissociation of pluripotent clusters and plated into a 9.6-cm2 well containing MEF feeders in K15F5 medium. Morphologically distinct mEpiSC colonies became evident over the next 4–8 days and were subcloned from a mixed population of cells, including mESCs. mEpiSC colonies were manually dissociated into small clusters using a glass needle and plated into 1.9-cm² wells containing MEF feeders in mEpiSC medium consisting of Knockout DMEM (Gibco, #10829018) supplemented with 20% Knockout Serum Replacement (Gibco, #10828010), 2 mM Glutamax™ (Gibco, #35050061), 1× nonessential amino acids (Gibco, #11140050), 0.1 mM β-mercaptoethanol (Sigma, #M7522), and 10-ng/ml FGF2 (R&D Systems, #233-FB).

After derivation, mEpiSCs were cultured in mEpiSC medium and passaged every third day using 1.5 mg/ml collagenase type IV (GIBCO, #17104-019) with pipetting into small clumps. mEpiSC medium used to culture CHIR-treated

mEpiSCs was supplemented with 3 μM CHIR99021 (CHIR) (Stemgent #04–0004). mEpiSCs generated for RNA FISH staining were cultured, permeabilized, and fixed on fibronectin-coated (15 μg/ml) (Sigma #F1141) coverslips.

**RNA fluorescence in situ hybridization (FISH) probe labeling & precipitation**. Human and mouse probes [XIST BAC (BAC PAC, RP13-183A17); ATRX BAC (BAC PAC, RP11-42M11); USP9X BAC (Invitrogen, CTD 3174G14); XIST fosmid (BAC PAC, G135P63425C4)] were labeled with Fluorescein-12-dUTP (Invitrogen), Cy3-dCTP (GE Healthcare, #PA53021), or Cy5-dCTP (GE Healthcare, #PA55031). Labeled probes for multiple genes were precipitated in a 3 M sodium acetate (Teknova, #S0298) solution along with 300 μg of yeast tRNA (Invitrogen, #15401–029), and 150 μg of sheared, boiled salmon sperm DNA (Invitrogen, #15632–011). The solution was then centrifuged at 21,130 X g for 20 min at 4 °C. The resulting pellet was washed in 70% ethanol, then washed in 100% ethanol, dried, and re-suspended in deionized formamide (ISC Bioexpress, #0606–500 ML). The re-suspended probe was denatured via incubation at 90 °C for 10 min followed by an immediate 5 min incubation on ice. A 2X hybridization solution consisting of 4X SSC, 20% Dextran sulfate (Millipore, #S4030), and 2.5 mg/ml purified BSA (New England Biolabs, #B9001S) was added to the denatured probe/formamide solution. Probes were stored at −20 °C until use.

**RNA FISH staining**. hESCs, differentiated mEpiLCs, and mEpiSCs grown on coverslips were permeabilized through sequential treatment with ice-cold cytoskeletal extraction (CSK) buffer containing 100 mM NaCl (ThermoFisher Scientific, #BP358), 300 mM sucrose (ThermoFisher Scientific, #BP220), 3 mM $MgCl_2$ (ThermoFisher Scientific, #AA12315A7), and 10 mM PIPES buffer (Sigma-Aldrich, #P6757), pH 6.8 for 30 s; ice-cold CSK buffer containing 0.4% Triton X-100 (Fisher Scientific, #EP151) for 30 s; followed twice with ice-cold CSK buffer for 30 s each. After permeabilization, cells were fixed by incubation in 4% paraformaldehyde (Electron Microscopy Sciences, #15710) for 10 min. Cells were then rinsed 3 times in 70% ethanol and stored in 70% ethanol at −20 °C prior to RNA FISH staining. Prior to RNA FISH probe hybridization, coverslips were dehydrated through 2 min incubations in 70%, 85%, 95%, and 100% ethanol solutions and subsequently air-dried for 15 min. The coverslips were then hybridized to the FISH probe overnight in a humid chamber at 37 °C. The samples were then washed 3 times for 7 min each at 37 °C with 2X SSC (Invitrogen, #AM9765)/50% deionized formamide (ISC Bioexpress, VWR:#0606), 2X SSC, and 1X SSC. A 1:250,000 dilution of DAPI (Invitrogen, #D21490) was added to the third 2X SSC wash. Coverslips were then mounted on slides in Vectashield (Vector Labs, #H-1000) and sealed with nail polish.

**Microscopy**. Coverslips containing stained cells were imaged using a Nikon Eclipse TiE inverted microscope with a Photometrics CCD camera. The images were deconvolved and uniformly processed using NIS-Elements software (Version 4.60.00).

**Quantification of RNA FISH stains**. Expression of X-linked genes was quantified at 10X to 100X resolution starting at the upper left corner of each image. Percent expression of monoallelic XIST RNA was calculated for each colony on the coverslip and recorded in one of the following expression categories for all experiments except Fig. 5: 0%, 1–19%, 20–39%, 40–59%, 60–79%, 80–99%, 100%. Figures 5, 6 display XIST RNA coating data in only three key expression categories (0–19%, 20–79%, 80–100%), as these categories most clearly demonstrated XIST RNA coating loss. Expression of X-linked genes ATRX and USP9X in individual nuclei was quantified in the following categories: nuclei with XIST RNA coats and monoallelic ATRX or USP9X expression; nuclei with XIST RNA coats and biallelic ATRX or USP9X expression; nuclei without XIST RNA coats and monoallelic ATRX or USP9X expression; and, nuclei without XIST RNA coats and biallelic ATRX or USP9X expression. hESC colonies with <100 cells or colonies with excessive cell overlaps were not quantified. All measurements were taken from distinct samples. Raw quantification data for all RNA FISH experiments in this study are included in Source Data file.

**RNA sequencing (RNA-Seq) sample preparation**. Total RNA was isolated from TRIzol (Life Technologies, #15596-018) according to the manufacturer's instructions. cDNA libraries were generated and sequenced on the Illumina HiSeq2000 platform to generate 75 bp single-end reads.

**Analysis of RNA-seq data**. Quality control analysis was conducted using FastQC (Version 0.11.9). Reads were aligned to the hg19 (human) reference genome using STAR[111] (Version 2.7.10a) and counted using FeatureCounts[112] (Version 1.22.2). Differential expression analysis was conducted using DESeq2 (Version 1.34.0).

**Materials and correspondence**. Requests for hESC lines should be directed to Gary D. Smith (smithgd@umich.edu). Requests for mESC and mEpiSC lines and other resources, reagents, or information should be directed to Sundeep Kalantry (kalantry@umich.edu).

**Statistics**. General linear model regression analysis was performed in R to determine statistically significant differences in XIST RNA coating through passaging (Figs. 1–10). For experiments comparing different culture conditions (Figs. 2–10), $p$-values for individual linear models or $p$-values comparing the linear models are reported, as indicated. A threshold of $p = 0.05$ was used to test for statistical significance. Principal Component Analysis (PCA) was conducted using R.

**Reporting summary**. Further information on research design is available in the Nature Research Reporting Summary linked to this article.

## Data availability

The data supporting the findings of this study are available from the corresponding authors upon reasonable request. All hESC lines generated for this study have been deposited to the NIH Human Embryonic Stem Cell Registry (https://grants.nih.gov/stem_cells/registry/current.htm). The raw and processed RNA-Seq data generated in this study have been deposited in the Gene Expression Omnibus under accession code GSE157809. Source data, which include raw RNA FISH quantification and differential expression datasets, are provided with this paper.

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

## Acknowledgements

We thank members of the Kalantry and Smith laboratories for discussions and critical review of the manuscript. This work was funded by NIH National Research Service Awards 5-T32-GM07544 (University of Michigan Predoctoral Genetics Training Program; to M.C.); T32-HD079342 (University of Michigan Predoctoral Career Training in the Reproductive Sciences Program; to M.C.); an NIH NIGMS R01 Award (R01GM124571) (to S. Kalantry); an NIH NICHD R01 Award (R01HD095463) (to S. Kalantry); a Reproductive Science Program Pilot Grant (to S. Ka. and G.D.S.); the University of Michigan Endowment for Basic Sciences (to S. Ka.); a University of Michigan Rackham Predoctoral Fellowship (to M.C.); MStem Cell Lab Funding (to G.D.S.); the University of Michigan President's Office (to G.D.S.); Michigan Medicine (to G.D.S.); the A. Alfred Taubman Medical Research Institute (to G.D.S.); the University of Michigan Department of Obstetrics and Gynecology (to G.D.S.); and, the American Society for Reproductive Medicine (ASRM) Research Institute (to G.D.S.). We also thank Dr. Mark R. Hughes of Genesis Genetics for performing early passage hESC karyotyping by aCGH.

## Author contributions

L.K., S.M.-P., I.E., and A.M.D.R. derived and cultured hESCs. M.C. performed mouse cell culture, RNA FISH, microscopic imaging and quantification of RNA FISH data, and molecular biology experiments. S. Ku., M.C., E.B., B.L., and A.W. performed RNA FISH of hESCs and quantified the RNA FISH data. M.C. conducted RNA-Seq analysis. K.C identified TCF binding sites near human *XIST* and mouse *Xist*. G.D.S., A.M.D.R., M.C., S. Ku., and S. Ka. designed experiments and analyzed data. M.C., S. Ku., G.D.S., and S. Ka. prepared the manuscript. All authors read and commented on the manuscript.

## Competing interests

The authors declare no competing interests.
