## [Peer Review File · Nature Communications]

Title: Preventing Erosion of X-chromosome Inactivation in Human Embryonic Stem CellsREVIEWER COMMENTS

Reviewer #1 (Remarks to the Author):

The authors derived several new female hESC lines to study potential effects of some variables during hESC culture on erosion of inactive X chromosome. They found that culturing hESCs on HFFs with xeno-free (XF) medium under 5% oxygen concentration stably maintained XIST expression but readily lost if XF medium was replaced with mTeSR1. They identified the presence of LiCl in the latter was a potential cause that produced such difference. Furthermore, they suspected that the effect of LiCl was inhibition of GSK3 pathway and experimentally showed that inhibition of GSK-3 pathway by drugs in the absence of Li recapitulated erosion.

This study suggests that GSK-3 pathway plays an important role in stable maintenance of X inactivation state in hESCs and inhibition of the pathway directly or indirectly causes erosion of the inactive X chromosome. Although the molecular basis of how GSK3 pathway exerts its effects on stable maintenance of X inactivation is still unknown, the finding would still be worth shearing among the people in the community of stem cell biology to improve the way to derive epigenetically stable female hPSCs. However, I still have some comments and questions as shown below.

I think that the authors need to describe more clearly how hESCs and their colonies look like in terms of the number, shape, density, etc. at the stage they refer to as passage 1 and the subsequent early passages. They started with isolated ICM cells and culture them on HFFs with XF medium under 5% O₂. I guess an epithelial sheet of outgrowth formed by the ICM cells as shown in Extended Data Figure 1 would then be dissociated or cut into small pieces to facilitate expansion of cell population. Is passage 1 the stage at which these split cells from the first epithelial sheet have grown to form some colonies? According to supplemental Table, they only examined 1-2 colonies for Xist expression at passage 1, and 3-4 colonies at passage 2. I am wondering if these colonies were all they had at the respective passage or a part of more colonies. Given my assumption that once an epithelial sheet was split, many colonies would arise at the next round of culture, the latter would be the case. If so, the numbers of colonies they examined would be too small. In addition, they did not provide the total number of nuclei they examined by RNA-FISH. If they examined a large number of nuclei, it is a little bit hard to believe that 100% of nuclei were positive for Xist-coating in Figure 1 and 3.

Did hESCs with stable XIST-coating in early passages contain only one XIST cloud? I am just wondering if naïve hESCs, in which both Xs are coated with XIST RNA, arise under this culture condition.

The importance of their finding that hESCs with stable Xist-coating can be derived using HFFs as feeder with XF medium under 5% oxygen would be strengthened if they showed that transcriptome of hESCs thus derived resembles that of epiblast cells in vivo, which should be available in public data base.

I am wondering if the effects of inhibiting the GSK3 pathway on erosion of X inactivation are unique to hESCs or also seen in other types of cells with an inactive X such as differentiated human somatic cells

and mouse EpiSCs. I am also curious if the inhibition of the GSK3 pathway prevents upregulation of Xist upon differentiation of mouse ESCs,

Reviewer #2 (Remarks to the Author):

The manuscript by Cloutier et al investigates an important problem, namely how the erosion of the inactive X-chromosome (Xi) arises in newly derived human primed ESCs and how this process can be prevented. Xi erosion affects nearly all female human primed ESC lines and leads to loss of Xist RNA expression and reactivation of a set of X-linked genes. Since Xi-erosion cannot be repaired upon differentiation due to permanent silencing of Xist, this process leads to dosage compensation issues in female primed hESCs. Consequently, all research employing female hESCs and hiPSCs might be negatively affected by Xi-erosion. Here, the authors uncover that the composition of culture media dramatically influences the expression of XIST, the key regulator of X-inactivation. Specifically, they show that hESCs cultured in defined medium stably maintains XIST expression and coating, whereas hESCs cultured in the widely used TeSR1 medium consistently lost XIST expression. The authors identify lithium chloride in the TeSR1 medium as a cause for XIST RNA loss as the addition of Lithium chloride to defined medium induces Xi-erosion. They then demonstrate that the inhibition of GSK-3 proteins (which are known to be inhibited by LiCl) has the same effect, suggesting that lithium chloride leads to Xi-erosion by activating Wnt signaling. For these experiments the authors derived various new hESC lines and analyzed expression of XIST across many passages. Overall, this paper is important for the community and provides a path forward to the prevention of Xi-erosion in typical primed hESC cultures. Thus, in my opinion, this paper definitely should be published in Nature Communications. Yet, I have several points that need to be addressed before publication, which I list below.

Major Points

The quantification of Xist expression per colony is interesting, yet, does not reveal the actual proportion of cells that is Xist-positive or Xist-negative. It is important to show the actual number of cells with Xist expression. Moreover, the classification of Xist-positive colonies changes between figures, for instance 7 classes in figures 1/3/4, 5 in figures 2/5 and 3 in figure 6. It is also unclear what exactly the relative cumulative frequency represents.

It is important to show more FISH images in the main figures, giving examples for the key results.

In the first figure, many passages were required before the complete loss of Xist can be observed. This observation raises the question of why other experiments were not taken similarly far (with respect to passages) to assess if erosion completes or stabilizes over time of passaging under other conditions. This is important as many experiments end with incomplete Xist loss.

Erosion of the Xi is defined by Xist loss and the reactivation of some genes on the X chromosome. There

are different arguments in the literature as to which event is first. Thus, determining X-linked gene silencing is critical for the demonstration that Xi erosion is rescued and needs to be provided for key experiments.

It has been argued that differentiation alters the X state, whereas other papers have suggested that the state of the X in primed hESCs is maintained upon differentiation. This is the feature of Xi-erosion that may most critically affect many researchers in the field. The authors therefore need to address this problem by performing differentiation assays and repeating the XiST quantifications and that of X-linked gene expression.

Can the authors provide any insights into the mechanisms that lead to Xist silencing via the regulation of signaling.

In Figure 3 the authors show that hESCs cultured on human feeder fibroblasts together with XF medium (?) do not lose XIST. Yet, they stop at p14 and at some passages describe a large fraction of colonies as 80-99% XIST positive – which could be the start of Xi-erosion. It would be nice to see more cell lines be analyzed, quantified at the per cell level and take out to longer passage. Regardless, it does appear that erosion cannot be completely blocked in the culture conditions described in Figure 3. Thus, the authors also need to be clear about whether they are completely blocking Xi erosion.

Does the freeze/thaw process affect the result in Figure 3. Since most researchers in the field will freeze and thaw their hESCs, it is important to address this question.

The methods and experimental details are not clear and should be improved upon. The figure legend should be clear on the culture conditions used for both the medium and the feeders/surface.

How does normoxia change the results on HFF/XF conditions, such as in Figures 4 or 6?

The change of scoring in Figure 6 makes it difficult to interpret the figure and understand if there is no erosion in 6a.

Are human feeders required or would mouse feeders yield the same result?

How can a cell line transition from being eroded (7f – passage 10/11) to being very little eroded (p12)?

Minor Points:

I assume cells for Figure 1 have been cultured in XF medium?

The images in figure S1 are overexposed or too enhanced (for Nanog and Oct4)

Figure S2 – expression is misspelled.

The Xist FISH images in Figure S2 are a little disappointing. In S2a – the Xist cloud looks very small on the left and enhanced on the left. The images in S2b are hard to read – could entire colonies be shown?

REVIEWER COMMENTS

We thank both reviewers for taking the time to review our manuscript in detail and providing very valuable feedback. We have endeavored to address all of the reviewer comments and provide our response below in blue font. The most significant changes to the manuscript are listed below:

- Testing the stability of X-inactivation in differentiating human embryonic stem cells (hESCs) (Fig. 7 in the revised manuscript).
- Evaluating the impact of GSK-3 inhibition on X-inactivation stability in mouse ESCs and epiblast stem cells (EpiSCs).
- Comparing the transcriptome via RNA sequencing of hESCs treated with GSK-3 inhibitors to untreated hESCs, naïve hESCs, and differentiated human tissue types.
- Identifying conserved DNA binding sequences of the Wnt effector TCF upstream of human and mouse *XIST/Xist* transcription start site (TSS).

Reviewer #1 (Remarks to the Author):

The authors derived several new female hESC lines to study potential effects of some variables during hESC culture on erosion of inactive X chromosome. They found that culturing hESCs on HFFs with xeno-free (XF) medium under 5% oxygen concentration stably maintained XIST expression but readily lost if XF medium was replaced with mTeSR1. They identified the presence of LiCl in the latter was a potential cause that produced such difference. Furthermore, they suspected that the effect of LiCl was inhibition of GSK3 pathway and experimentally showed that inhibition of GSK-3 pathway by drugs in the absence of Li recapitulated erosion.

This study suggests that GSK-3 pathway plays an important role in stable maintenance of X inactivation state in hESCs and inhibition of the pathway directly or indirectly causes erosion of the inactive X chromosome. Although the molecular basis of how GSK3 pathway exerts its effects on stable maintenance of X inactivation is still unknown, the finding would still be worth shearing among the people in the community of stem cell biology to improve the way to derive epigenetically stable female hPSCs. However, I still have some comments and questions as shown below.

I think that the authors need to describe more clearly how hESCs and their colonies look like in terms of the number, shape, density, etc. at the stage they refer to as passage 1 and the subsequent early passages. They started with isolated ICM cells and culture them on HFFs with XF medium under 5% O₂. I guess an epithelial sheet of outgrowth formed by the ICM cells as shown in Extended Data Figure 1 would then be dissociated or cut into small pieces to facilitate expansion of cell population. Is passage 1 the stage at which these split cells from the first epithelial sheet have grown to form some colonies?

We now provide a better description in the Methods section of how the hESC lines were derived and passaged. Passage 1 represents the first passage generated by physical/mechanical/manual splitting of the ICM outgrowth (referenced “sheet”). We now also

include additional representative images in Supplementary Fig. 1a to show the 'passage 0' ICM and outgrowth and 'passage 1' hESCs.

According to supplemental Table, they only examined 1-2 colonies for Xist expression at passage 1, and 3-4 colonies at passage 2. I am wondering if these colonies were all they had at the respective passage or a part of more colonies. Given my assumption that once an epithelial sheet was split, many colonies would arise at the next round of culture, the latter would be the case. If so, the numbers of colonies they examined would be too small.

Yes, we examined all available colonies at each passage. In the first few passages, though, there were only a few colonies that could be expanded by passaging. Furthermore, only a small subset of the cells at each early passage could be set aside for RNA FISH staining. The remainder of the cells at these early passages were used for further expansion in order to freeze the cells at early passages and also to continue culturing the cells for analyses at subsequent passages. Once the hESC lines were established and cryo-preserved, typically by passage (P) 5, the number of colonies for RNA FISH analyses increased, for example, in Fig. 1 to 39 colonies.

In addition, they did not provide the total number of nuclei they examined by RNA-FISH. If they examined a large number of nuclei, it is a little bit hard to believe that 100% of nuclei were positive for Xist-coating in Figure 1 and 3.

We apologize for omitting the number of nuclei we counted in our RNA FISH stains in the original submission. In the revised manuscript, we now include the number of cells analyzed/colony as part of a detailed description of our analysis method in the first paragraph of the results section. In all of our RNA FISH measurements, we counted a minimum of 100 cells in each colony, even at the earliest passages with few colonies. Indeed, all of the nuclei in all of the colonies were *XIST* RNA-coated in the initial passages in Figs. 1 and 3.

Relatedly, we wish to clarify the reason we segmented the data into the percentage of *XIST* RNA-coated nuclei per colony rather than as total numbers of nuclei with or without *XIST* RNA coats. The reason is because of the manner in which the hESC colonies are passaged. As the reviewer noted earlier, each colony is manually cut with glass capillaries and passaged. Since the colonies are not dissociated into single cells during passaging, each colony that arises after passaging is therefore derived from a group of cells rather than from a single cell. Presenting the data as numbers of nuclei with or without *XIST* RNA coats per passage could lead to a biased representation of the data. For example, a large hESC colony that upon passaging also yields a large hESC colony could bias the total proportion of nuclei with or without *XIST* RNA coats. Since each colony is clonally derived from another colony and not from a single cell, segmenting the data into percentage of nuclei with *XIST* RNA coats per colony rather than aggregating all the nuclei with or without *XIST* RNA coats more accurately represents the data. We therefore plot the percentages of nuclei with *XIST* RNA coats per colony for all the colonies in all RNA FISH stains. Furthermore, our graphs show the proportion of colonies in a given passage that harbor each level of *XIST* RNA coating. These data provide a comprehensive view of *XIST* RNA coating at each passage of the hESC lines analyzed. In the revised manuscript, we discuss how the hESCs were passaged and our RNA FISH counting strategy in the first paragraph of the Results section (p. 6, lines 94-103). We also edited the labels in the graphs to make them simpler and more intuitive.

Did hESCs with stable XIST-coating in early passages contain only one XIST cloud? I am just wondering if naïve hESCs, in which both Xs are coated with XIST RNA, arise under this culture condition.

The hESCs at the earliest passages either entirely or predominantly contained one *XIST* RNA coat. We did not observe hESCs with two *XIST* RNA coats in any of our culture conditions, including with the mTeSR1 medium, XF medium, and in the XF medium with LiCl or GSK-3 inhibitors. Two *XIST* RNA coats in the nucleus is a hallmark of naïve female hESCs, which require a cocktail of inhibitors to cellular signaling pathways in order to derive. Our hESC culture conditions do not employ these inhibitors and are consistent with the generation of primed pluripotent hESCs, which are characterized either with a single *XIST* RNA coat or, upon X-inactivation erosion, no *XIST* RNA coat.

The importance of their finding that hESCs with stable Xist-coating can be derived using HFFs as feeder with XF medium under 5% oxygen would be strengthened if they showed that transcriptome of hESCs thus derived resembles that of epiblast cells in vivo, which should be available in public data base.

As suggested by the reviewer, we performed RNA-Seq on hESC line 90-14 cultured under different conditions and compared them to published RNA-Seq datasets from other hESC lines, human epiblast cells from blastocyst-stage embryos, and human kidney and liver tissues. We sequenced or mined data from the following samples:

- hESCs cultured in HFFs in XF medium with 5% O₂.
- hESCs cultured with the mTeSR medium in 5% O₂.
- hESCs cultured with XF medium + LiCl in 5% O₂.
- hESCs cultured with XF medium + Ly209 (a GSK-3 inhibitor) in 5% O₂.
- hESCs cultured with XF medium + BIO (a GSK-3 inhibitor) in 5% O₂.
- hESC RNA-Seq data from Patel et al. (2017).
- hESC RNA-Seq data from Sun et al. (2018).
- Epiblast cells from human blastocyst-stage embryos (Petropoulos et al., 2018).
- Human kidney (Liao et al., 2020).
- Human liver (MacParland et al., 2018).

A relative comparison of these data is presented in a principal component (PC) plot in Supplementary Fig. 5.

I am wondering if the effects of inhibiting the GSK3 pathway on erosion of X inactivation are unique to hESCs or also seen in other types of cells with an inactive X such as differentiated human somatic cells and mouse EpiSCs.

This is a good question. As suggested by the reviewer, in the revised manuscript we tested X-inactivation erosion in mouse (m) EpiSCs by culturing the mEpiSCs with the GSK-3 inhibitor CHIR99021 (CHIR). CHIR is part of the '2i' culture medium used to culture naïve mouse (m) ESCs. mEpiSCs cultured with CHIR at the concentration (3 μM) used to culture naïve mESCs also lost *Xist* RNA coating. These data are included in Fig. 10 and discussed in the manuscript on p. 15-16, lines 296-303. The impact of GSK-3 inhibition, therefore, appears to be conserved between human and mouse primed pluripotent stem cells.

I am also curious if the inhibition of the GSK3 pathway prevents upregulation of Xist upon differentiation of mouse ESCs.

As suggested by the reviewer, we differentiated mESCs into epiblast-like cells (mEpiLCs) and then cultured the mEpiLCs with media containing 3 μ M CHIR99021. Like the mEpiSCs cultured with 3 μ M CHIR, the mEpiLCs cultured with CHIR also lost *Xist* RNA coating. These data are included in Fig. 9 and discussed in the manuscript on p. 15, lines 283-295.

Reviewer #2 (Remarks to the Author):

The manuscript by Cloutier et al investigates an important problem, namely how the erosion of the inactive X-chromosome (Xi) arises in newly derived human primed ESCs and how this process can be prevented. Xi erosion affects nearly all female human primed ESC lines and leads to loss of Xist RNA expression and reactivation of a set of X-linked genes. Since Xi-erosion cannot be repaired upon differentiation due to permanent silencing of Xist, this process leads to dosage compensation issues in female primed hESCs. Consequently, all research employing female hESCs and hiPSCs might be negatively affected by Xi-erosion. Here, the authors uncover that the composition of culture media dramatically influences the expression of XIST, the key regulator of X-inactivation. Specifically, they show that hESCs cultured in defined medium stably maintains XIST expression and coating, whereas hESCs cultured in the widely used TeSR1 medium consistently lost XIST expression. The authors identify lithium chloride in the TeSR1 medium as a cause for XIST RNA loss as the addition of Lithium chloride to defined medium induces Xi-erosion. They then demonstrate that the inhibition of GSK-3 proteins (which are known to be inhibited by liCl) has the same effect, suggesting that lithium chloride leads to Xi-erosion by activating Wnt signaling. For these experiments the authors derived various new hESC lines and analyzed expression of XIST across many passages. Overall, this paper is important for the community and provides a path forward to the prevention of Xi-erosion in typical primed hESC cultures. Thus, in my opinion, this paper definitely should be published in Nature Communications. Yet, I have several points that need to be addressed before publication, which I list below.

Major Points

The quantification of Xist expression per colony is interesting, yet, does not reveal the actual proportion of cells that is Xist-positive or Xist-negative. It is important to show the actual number of cells with Xist expression. Moreover, the classification of Xist-positive colonies changes between figures, for instance 7 classes in figures 1/3/4, 5 in figures 2/5 and 3 in figure 6. It is also unclear what exactly the relative cumulative frequency represents.

We apologize for omitting the number of nuclei we counted in our RNA FISH stains in our original submission. We now include the number of cells analyzed/colony as part of a detailed description of our analysis method in the first paragraph of the results section. In all of our RNA FISH measurements, we counted a minimum of 100 cells in each colony, even at the earliest passages with few colonies. Indeed, all of the nuclei in all of the colonies were *XIST* RNA-coated in the initial passages in.

Relatedly, we wish to clarify the reason we segmented the data into the percentage of *XIST* RNA-coated nuclei per colony rather than as numbers of nuclei with and without *XIST* RNA

coats. It is because of the manner in which the hESC colonies are passaged. Each colony is manually cut with glass capillaries and passaged. Since the colonies are not dissociated into single cells during passaging, each colony that arises after passaging is therefore derived from a group of cells rather than from a single cell. Presenting the data as numbers of nuclei with or without *XIST* RNA coats per passage could lead to a biased representation of the data. For example, a large hESC colony that upon passaging also yields a large hESC colony could bias the total proportion of nuclei with or without *XIST* RNA coats. Since each colony is clonally derived from another colony and not from a single cell, segmenting the data into percentage of nuclei with *XIST* RNA coats per colony rather than aggregating all the nuclei with or without *XIST* RNA coats more accurately represents the data. We therefore plot the percentages of nuclei with *XIST* RNA coats per colony for all the colonies in all RNA FISH stains. Furthermore, our graphs show the proportion of colonies in a given passage that harbor each level of *XIST* RNA coating. These data provide a comprehensive view of *XIST* RNA coating at each passage of the hESC lines analyzed. In the revised manuscript, we discuss how the hESCs were passaged and our RNA FISH counting strategy in the first paragraph of the Results section (p. 6, lines 94-103). We also edited the labels in the graphs to make them simpler and more intuitive.

As for the variable representation of the data, we have now changed the plotting so that all figures other than Figs. 5 and 6 have seven categories. In Figs. 5 and 6, to simplify and make the data clearer to understand, we reduced the seven categories into the three major categories that encapsulate the seven categories. These three categories comprise the key categories in the experiment: colonies with 80-100% of the nuclei with *XIST* RNA coats; 20-79% of the nuclei with *XIST* RNA coats; and, 0-19% of the nuclei with *XIST* RNA coats. As a comparison, we have included the Fig. 5 data subdivided into seven categories in Supplementary Fig. 4. In Fig. 6, the data demonstrate that culturing the hESCs with XF medium maintains *XIST* RNA coating in 80-100% of the nuclei in all of the colonies assayed across all of the passages from P3-19 (Fig. 6a). By contrast, when the hESCs are cultured with XF medium supplemented with LiCl or with mTeSR medium, more than half of the colonies lack any nuclei with *XIST* RNA coating (Fig. 6b-c). Whereas the frequency of *XIST* RNA coating is not statistically significantly different across passages (general linear model, $p = 0.2$), the data in Fig. 6b-c are significantly different across passages (general linear model, $p < 0.001$). That 80-100% of the nuclei exhibit *XIST* RNA coating in hESCs cultured in the XF medium even at P19 argues against X-inactivation erosion in these cells.

It is important to show more FISH images in the main figures, giving examples for the key results.

We now include more RNA FISH stain images in the main figures to show representative colonies and nuclei. Figure 1b shows representative images of individual hESC nuclei in the following categories: a nucleus with *XIST* RNA coat and monoallelic expression of *ATRX* and *USP9X*; a nucleus with *XIST* RNA coat, monoallelic expression of *ATRX*, and biallelic expression of *USP9X*; a nucleus without *XIST* RNA coat and monoallelic expression of *ATRX* and *USP9X*; and, a nucleus without *XIST* RNA coat, monoallelic expression of *ATRX*, and biallelic expression of *USP9X*. Figure 1c shows representative hESC colonies with high, intermediate, and low numbers of nuclei with *XIST* RNA coats, in combination with *ATRX* and *USP9X* expression. Figure 9b shows representative mEpiLC colonies expressing high, intermediate, and low numbers of nuclei with *Xist* RNA coats. Figure 10b shows representative mEpiSC colonies expressing high, intermediate, and low numbers of nuclei with *Xist* RNA coats.

In the first figure, many passages were required before the complete loss of Xist can be observed. This observation raises the question of why other experiments were not taken similarly far (with respect to passages) to assess if erosion completes or stabilizes over time of passaging under other conditions. This is important as many experiments end with incomplete Xist loss.

In Fig. 1d, as proof-of-concept, we show that hESC line UM33-4 loses *XIST* RNA coating during passaging and that by P27 all of the hESC colonies have lost *XIST* RNA coating in all of the nuclei. In subsequent experiments, our goal was to test and compare whether a similar trend of *XIST* RNA coating loss would occur when hESCs were cultured under different conditions. We therefore used Fig. 1d to determine which passages experienced *XIST* RNA coating loss and focused our subsequent analyses on these passages. Our experimental goal was not to demonstrate complete loss of *XIST* RNA coating in the subsequent experiments. Moreover, we statistically compared the trends across passages in a given culture condition and calculated significance of any loss of *XIST* RNA coating across passages even in the absence of complete loss of *XIST* RNA coating.

Erosion of the Xi is defined by Xist loss and the reactivation of some genes on the X chromosome. There are different arguments in the literature as to which event is first. Thus, determining X-linked gene silencing is critical for the demonstration that Xi erosion is rescued and needs to be provided for key experiments.

This is again a good point. We quantified expression of the X-linked genes *ATRX* and *USP9X* by RNA FISH in cells with and without *XIST* RNA coating in the UM 63-1 and UM 77-2 cultured in mTeSR1 medium across passages (Supplementary Fig. 2). The trends suggest that beginning at P5 in cells with *XIST* RNA coats, *USP9X* is expressed monoallelically in most hESCs. In hESCs without *XIST* RNA coats, though, *USP9X* is detected biallelically in many hESCs. *ATRX* expression, however, remains monoallelic both in cells with and without *XIST* RNA coats. *USP9X* biallelism occurs predominantly in the absence of *XIST*. From these data, we infer that biallelism of X-linked genes follows loss of *XIST* RNA coating. Our data also supports a gradual re-expression of X-linked genes during X-inactivation erosion, which is consistent with other published reports (Shen et al., 2008; Dvash et al., 2010; Lengner et al., 2010; Mekhoubad et al., 2012; Anguera et al., 2012; Vallot et al., 2015; Xie et al., 2016).

It has been argued that differentiation alters the X state, whereas other papers have suggested that the state of the X in primed hESCs is maintained upon differentiation. This is the feature of Xi-erosion that may most critically affect many researchers in the field. The authors therefore need to address this problem by performing differentiation assays and repeating the XiST quantifications and that of X-linked gene expression.

As suggested by the reviewer, we undertook a series of experiments to differentiate hESCs and measure *XIST* RNA coating to test the stability of X-inactivation during hESC differentiation. We differentiated two different hESC lines (UM63-1 and UM77-2) cultured with XF medium, which maintains *XIST* RNA coating in undifferentiated hESCs, into embryoid bodies (EB) under three different culture conditions (described below) and assayed *XIST* RNA coating by RNA FISH at 3 days (d), 6d, and 9d of differentiation.

We first differentiated the hESC lines with a commercial hESC differentiation medium, AggreWell, which resulted in the progressive loss of *XIST* RNA coating with increasing days of EB differentiation. Upon enquiring with the manufacturer of AggreWell medium, we discovered that AggreWell is based on the mTeSR1 medium that contains lithium chloride (LiCl). Secondly, we differentiated the hESCs into EBs with XF medium lacking the bFGF growth factor that promotes pluripotency and is normally a part of the XF hESC culture medium. Thirdly, we differentiated the hESCs into EBs in XF medium without bFGF but with LiCl added at the same concentration as in the AggreWell medium (0.98 mM). Whereas the use of the XF medium without bFGF2 led to some but not significant loss of *XIST* RNA coating in the EBs, the addition of LiCl to the XF medium without bFGF led to increased loss of *XIST* RNA coating, approaching the pattern observed in the EBs differentiated with the AggreWell medium. These data again reinforce that LiCl causes loss of *XIST* RNA coating. These data also suggest that hESCs can undergo slight X-inactivation erosion even when the hESCs are differentiated without LiCl. These data are included in Fig. 7 and discussed in the text on p.12-13 (lines 233-251).

Can the authors provide any insights into the mechanisms that lead to *Xist* silencing via the regulation of signaling.

We now include an expanded discussion of how Wnt signaling could impact *XIST* RNA expression on pages 16-17 of the revised manuscript. A primary mode of action of GSK-3 β is the negative regulation of the canonical Wnt signaling pathway. Wnt signaling conventionally activates transcription through the binding of TCF and β -catenin to target sequences, but recent work has found that Wnt signaling can also lead to transcriptional repression in some contexts (Blauwkamp et al., 2008; Zhang et al., 2014). TCF binding at target sites in conjunction with the binding of cofactors may repress transcription.

To test whether Wnt signaling could directly regulate *XIST*, we examined human and mouse genomic sequence 5' of the *XIST/Xist* transcription start sites (TSSs) and found three conserved putative TCF binding motifs within this region (Fig. 10d). We also found that sequences surrounding these motifs are also conserved between humans and mice, suggesting that these sequences adjacent to TCF binding motifs may serve as platforms for the binding of other transcription factors that may in turn contribute to the silencing of *XIST/Xist* in the two species.

GSK-3 β and other components of the canonical Wnt signaling pathway also crosstalk with a number of other signaling pathways that may indirectly repress *XIST*. Thus, activation of the canonical Wnt pathway by GSK-3 β inhibition may either directly or indirectly repress *XIST* expression in human and mouse ESCs. Future work will dissect the direct vs. indirect regulation of *XIST* RNA expression by GSK-3 and Wnt signaling in these cells.

In Figure 3 the authors show that hESCs cultured on human feeder fibroblasts together with XF medium (?) do not lose *XIST*. Yet, they stop at p14 and at some passages describe a large fraction of colonies as 80-99% *XIST* positive – which could be the start of Xi-erosion. It would be nice to see more cell lines be analyzed, quantified at the per cell level and take out to longer passage. Regardless, it does appear that erosion cannot be completely blocked in the culture conditions described in Figure 3. Thus, the authors also need to be clear about whether they are completely blocking Xi erosion.

At P14 in Fig. 3, ~70% of the hESC colonies display 100% of their cells with *XIST* RNA coating. Nearly all of the remaining colonies display 80-99% of their nuclei with *XIST* RNA coating. A

statistical analysis of the data in Fig. 3 demonstrates a lack of significant change in the percentage of *XIST* RNA-coated nuclei/colony across passaging when the hESCs are cultured with the XF medium.

As for XF medium, it refers to the 'XenoFree' hESC culture medium (defined in the Results section of the manuscript). The XF medium chemical composition is included in the Methods section.

Does the freeze/thaw process affect the result in Fig. 3. Since most researchers in the field will freeze and thaw their hESCs, it is important to address this question.

In Figs. 3 and 4, the cells were not frozen and thawed. In Figs. 5 and 7, though, the hESCs were frozen and thawed. We speculate that the stress of freeze-thaw may subtly influence *XIST* RNA expression in hESCs, as noted in the manuscript. Though, we respectfully feel that addressing this issue rigorously is beyond the scope of this manuscript.

The methods and experimental details are not clear and should be improved upon. The figure legend should be clear on the culture conditions used for both the medium and the feeders/surface.

Thank you. We have significantly edited the Methods, Figures, and the Figure legends to make them clearer.

How does normoxia change the results on HFF/XF conditions, such as in Figures 4 or 6?

We comprehensively investigated hESC culture in normoxic conditions (20% O₂) with two hESC lines (UM63-1 and UM77-2) in Fig. 3, where we excluded normoxia as a cause of *XIST* RNA expression loss. Therefore, respectfully, we didn't feel it was necessary to repeat the same experiment with these two lines in Fig. 4 or with the UM90-14 hESC line in Fig. 6.

The change of scoring in Figure 6 makes it difficult to interpret the figure and understand if there is no erosion in 6a.

Please see the earlier discussion on this issue. A statistical analysis of the data demonstrates that the pattern of *XIST* RNA coated nuclei/nuclei across passaging in Fig. 6a does not change significantly, but the pattern does change significantly in Fig. 6b-c.

Are human feeders required or would mouse feeders yield the same result?

We, unfortunately, did not derive hESCs on mouse fibroblast feeder cells. The reason to derive hESCs on human fibroblast feeders (HFFs) is that it allows for xenofree derivation and culture conditions – the hESCs and HFFs are from the same species.

How can a cell line transition from being eroded (7f – passage 10/11) to being very little eroded (p12)?

Although the same hESC line is investigated across passages, a different sample of cells is stained at each passage. In other words, the cells stained at each passage are different than the cells that are passaged to propagate the hESC line. Variability as observed between

P10/11 and P12 in Fig. 7f (now Fig. 8f in the revised manuscript) and between other passages in other figures is not unusual, as each RNA FISH stain is a snapshot of the entire cell population, which may change from passage to passage. In each experiment, we calculated the significance of any changes across passages in the percentage of nuclei with *XIST* RNA coating per colony.

Minor Points:

I assume cells for Figure 1 have been cultured in XF medium?

Thank you. We have updated Fig. 1 (and the other Figures) to indicate which passages were cultured in XF medium and which were cultured in mTeSR1 medium.

The images in figure S1 are overexposed or too enhanced (for Nanog and Oct4).

We now provide other stain images detecting OCT4 and NANOG in Supplementary Fig. 1b.

Figure S2 – expression is misspelled.

Thank you. We have now fixed this error.

The Xist FISH images in Figure S2 are a little disappointing. In S2a – the Xist cloud looks very small on the left and enhanced on the left. The images in S2b are hard to read – could entire colonies be shown?

We now provide alternate representative images of individual nuclei in Fig. 1b and of representative colonies in Fig. 1c.

REVIEWERS' COMMENTS

Reviewer #1 (Remarks to the Author):

I am satisfied with alterations the authors have made in the revised manuscript. I would like to, however, suggest the authors to include some more information from the data of RNA-seq analysis. The authors showed PCA to compare the transcriptome of hESCs cultured in different conditions and concluded that they are similar but not identical to one another. I think it would be nice if they provide the data about differentially expressed genes between hESCs cultured in XF, which do not undergo erosion, and those cultured in XF with LiCl or other GSK3 inhibitors, which undergo erosion.

Reviewer #2 (Remarks to the Author):

The authors did well with their revision and added new data based on the referee comments. I have no further comments and believe the paper will be important for the field.

REVIEWERS' COMMENTS

We thank both reviewers for taking the time to review our manuscript in detail and providing very positive and valuable feedback. We have endeavored to address each of the reviewer comments and provide our response below in blue font.

Reviewer #1 (Remarks to the Author):

I am satisfied with alterations the authors have made in the revised manuscript. I would like to, however, suggest the authors to include some more information from the data of RNA-seq analysis. The authors showed PCA to compare the transcriptome of hESCs cultured in different conditions and concluded that they are similar but not identical to one another. I think it would be nice if they provide the data about differentially expressed genes between hESCs cultured in XF, which do not undergo erosion, and those cultured in XF with LiCl or other GSK3 inhibitors, which undergo erosion.

Thank you for this feedback. We now provide lists in the Source Data file of genes that are differentially expressed in hESCs cultured in XF, mTeSR1, XF + LiCl, XF + BIO, and XF + Ly209 media. We only sequenced one replicate per culture condition and as a result the adjusted p-values for differentially expressed genes do not appear significant. We have also deposited our raw and processed RNA-Seq datasets in the Gene Expression Omnibus with accession number GSE157809.

Reviewer #2 (Remarks to the Author):

The authors did well with their revision and added new data based on the referee comments. I have no further comments and believe the paper will be important for the field.

Thank you for this feedback.